# LEARNABILITY OF CONVOLUTIONAL NEURAL NETWORKS FOR INFINITE DIMENSIONAL INPUT VIA MIXED AND ANISOTROPIC SMOOTHNESS

**Sho Okumoto**[1,†]**, Taiji Suzuki**[1,2,‡]

[1]Graduate School of Information Science and Technology, the University of Tokyo
[2]RIKEN Center for Advanced Intelligence Project
[†]`lebesgue0118@gmail.com`, [‡]`taiji@mist.i.u-tokyo.ac.jp`

## ABSTRACT

Among a wide range of success of deep learning, convolutional neural networks have been extensively utilized in several tasks such as speech recognition, image processing, and natural language processing, which require inputs with large dimensions. Several studies have investigated function estimation capability of deep learning, but most of them have assumed that the dimensionality of the input is much smaller than the sample size. However, for typical data in applications such as those handled by the convolutional neural networks described above, the dimensionality of inputs is relatively high or even infinite. In this paper, we investigate the approximation and estimation errors of the (dilated) convolutional neural networks when the input is infinite dimensional. Although the approximation and estimation errors of neural networks are affected by the curse of dimensionality in the existing analyses for typical function spaces such as the Hölder and Besov spaces, we show that, by considering anisotropic smoothness, they can alleviate exponential dependency on the dimensionality but they only depend on the smoothness of the target functions. Our theoretical analysis supports the great practical success of convolutional networks. Furthermore, we show that the dilated convolution is advantageous when the smoothness of the target function has a sparse structure.

## 1 INTRODUCTION

Deep learning has shown high performance in several tasks such as image recognition, speech recognition, and natural language processing. In particular, convolutional neural networks (CNNs) and dilated CNNs have been quite effective in tasks involving high-dimensional data (van den Oord et al., 2016; He et al., 2016; Simonyan & Zisserman, 2015; Yoon, 2014). However, many aspects of its theoretical nature are still unclear while related theoretical studies have attracted much attention. Aside from the analysis of CNNs, one of the most fundamental issues in deep learning theories is its function approximation and estimation capabilities. For example, it is well known that any continuous function with compact support can be approximated with arbitrary accuracy by a two-layer fully connected neural network (Cybenko, 1989; Hornik, 1991). Moreover, the representation ability of deep learning to approximate a function in some function classes such as Hölder classes has also been extensively analyzed (Mhaskar & Micchelli, 1992; Mhaskar, 1993; Chui et al., 1994; Mhaskar, 1996; Pinkus, 1999; Yarotsky, 2017; Petersen & Voigtlaender, 2017). In addition to the approximation ability, the estimation ability of deep learning for estimating a function by a finite sample has also been extensively studied. For example, Schmidt-Hieber (2020) derived the estimation error bound of deep learning with ReLU activation (Nair & Hinton, 2010; Glorot et al., 2011) to estimate functions in the Hölder space and showed the rate of convergence achieves the (near) minimax optimal rate. Suzuki (2019) derived approximation and estimation error rates of deep learning with ReLU activation for the Besov spaces, which were also shown to be (near) minimax optimal. Although the derived rates of convergence are near optimal, these studies assumed that the dimensionality of inputs is fixed and much smaller than the sample size. Indeed, the derived rates suffer from the curse of dimensionality. However, in practice, we often encounter settings where the input

dimensionality is larger than the sample size or even infinite. For example, in image recognition and natural language processing, the dimensionality of inputs (images or texts) is very large, and they could be seen as almost infinite dimensional.

To address this issue, some researches considered a setting where the dimensionality of the support of the data distribution is low dimensional. Chen et al. (2019b;a) considered a setting where data can be embedded in a low dimensional sub-manifold and derived the approximation error of functions that depends merely on the dimensionality of the sub-manifold instead of that of the entire space. Nakada & Imaizumi (2020) also considered a similar setting, and showed that the estimation error is characterized by the Minkowski dimension of the support of the data distribution. Suzuki (2019) showed that, even if the data cannot be embedded in a low dimensional manifold, anisotropic smoothness of the target function can mitigate the curse of dimensionality. Although these studies revealed that deep learning can avoid curse of dimensionality by utilizing some low dimensional structures of data and the target functions, it still remains unclear how deep learning performs for very high dimensional settings including an infinite dimensional setting. See Table 1 for a summarized comparison to existing studies.

In terms of infinite dimensional inputs, there have been already several studies on approximation and estimation errors for non-deep-learning methods. For example, so called hyperbolic cross approximation has been considered to approximate a function in a tensor product space with support on $[0,1]^\infty$ (Dũng & Griebel, 2016) and a polynomial order approximation is possible for functions with mixed smoothness, that is, specific summability properties of the smoothness indices are fulfilled. Ingster & Stepanova (2011) analyzed a Gaussian white noise model with an infinite dimensional input and showed that the estimation accuracy for signals on infinite dimensional anisotropic Sobolev spaces depends on the reciprocal sum of the smoothness per axis (see also Ingster & Stepanova (2006); Ingster & Suslina (2007); Ingster & Stepanova (2009)). Oliva et al. (2013; 2015) proposed methods to estimate a map where the input and output are functions or distributions, and derive the rate of convergence. Ferraty et al. (2007) analyzed the Nadaraya-Watson estimator when the inputs are functions, derived the convergence rate of the estimator, and gave the asymptotic confidence band in the context of functional data analysis (see Ling & Vieu (2018) as a comprehensive survey of the nonlinear functional data analysis literature). However, these researches are not for the deep learning and the benefit of deep learning for such situation has not been well characterized in the literature.

In this study, we analyze the approximation and estimation accuracy in a setting where the input is infinite dimensional, and derive their convergence rates. We assume that the true function has mixed and anisotropic smoothness, that is, the function has different smoothness toward different coordinate similarly to Dũng & Griebel (2016); Ingster & Stepanova (2011). The intuition behind this setting is as follows: Considering a function which takes an image as an input, an image can be decomposed into different frequency components and usually a function of images has less sensitivity on the high frequency components and more dependent on the low frequency components, which can be formulated as non-uniform smoothness toward each coordinate direction. By considering such a setting, we can show that the rate of convergence can avoid the curse of dimensionality and be of polynomial order. Our contribution can be summarized as follows:

1. We consider a learning problem in which the target function to be approximated or estimated can take an infinite dimensional input and has mixed or anisotropic smoothness. We show that deep learning by fully connected neural networks can achieve approximation and estimation errors dependent only on smoothness of the target function and independent of the dimension.

2. We also consider a setting where the smoothness of the target function has a sparse structure, and then we show that dilated CNNs can find appropriate variables and improve the rate of convergence. This indicates that CNNs can capture a long range dependence among the input.

These results show that even when the dimension $d$ of the data is very large compared to the number of observations $n$, or even when the input is infinite dimensional, it is possible to derive a polynomial order estimation error bound that depends only on the smoothness of the function class. This analysis partially explains the great success of CNNs in various applications with high dimensional inputs.

## 2 PROBLEM SETTING AND NOTATIONS

In this section, we prepare the notations and introduce the problem setting. Throughout this paper, we use the following notations. Let $\mathbb{R}_{>0} := \{s \in \mathbb{R} : s > 0\}$, and for a set $\mathbb{D}$, let $\mathbb{D}^\infty := \{(s_1, \ldots, s_i, \ldots) : s_i \in \mathbb{D}\}$ (for example, $\mathbb{R}^\infty := \{(s_i)_{i=1}^\infty : s_i \in \mathbb{R} \ (\forall i = 1, 2, \ldots)\}$). For

Table 1: Comparison of this work and existing work on theoretical analyses of deep learning for high dimensional data. $a = (a_i)_{i=1}^{\infty}$ is a smoothness parameter, $\tilde{a} := (\sum_{i=1}^{\infty} a_i^{-1})^{-1}$, $v = (1/p - 1/2)_+$, $s = a_1 = \cdots = a_d$ and $D$ is the dimensionality of low dimensional structure.

| Function class | mixed smooth ($d \ll n$) | anisotropic smooth ($d \ll n$) | low-dim data |
|---|---|---|---|
| Author | Suzuki (2019) | Suzuki & Nitanda (2021) | Nakada & Imaizumi (2020); Schmidt-Hieber (2019); Chen et al. (2019b;a) |
| Rate | $(n/\log(n)^{d-1})^{-\frac{2s}{2s+1}}$ | $n^{-\frac{2\tilde{a}}{2\tilde{a}+1}}$ | $n^{-\frac{2s}{2s+D}}$ |

| Function class | mixed smooth ($d = \infty$) | anisotropic smooth ($d = \infty$) |
|---|---|---|
| Author | This work | This work |
| Rate | $n^{-\frac{2(a_1-v)}{2(a_1-v)+1}}$ | $n^{-\frac{2(\tilde{a}-v)}{2(\tilde{a}-v)+1}}$ |

$s \in \mathbb{R}^{\infty}$, let $\text{supp}(s) = \{i \in \mathbb{N} : s_i \neq 0\}$. Let $\mathbb{N}_0^{\infty} := \{l \in (\mathbb{N} \cup \{0\})^{\infty} : \text{supp}(l) < \infty\}$ and define $\mathbb{Z}_0^{\infty}$ and $\mathbb{R}_0^{\infty}$ in the same way. Furthermore, for $s \in \mathbb{R}_0^{\infty}$, let $2^s := 2^{\sum_{i=1}^{\infty} s_i}$. For $L \in \mathbb{N}$, let $[L] = \{1, \ldots, L\}$. For $a \in \mathbb{R}$, let $\lfloor a \rfloor$ be the largest integer less than or equal to $a$.

## 2.1 REGRESSION PROBLEM WITH INFINITE DIMENSIONAL PREDICTOR

In this paper, we consider a regression problem where the predictor (input) is infinite dimensional. Let $\lambda$ be the uniform probability measure on $([0,1], \mathcal{B}([0,1]))$ where $\mathcal{B}([0,1])$ is the Borel $\sigma$-field on $[0,1]$, and let $\lambda^{\infty}$ be the product measure of $\lambda$ on $([0,1]^{\infty}, \mathcal{B}([0,1]^{\infty}))$ where $\mathcal{B}([0,1]^{\infty})$ is the product $\sigma$-algebra generated by the cylindric sets $\cap_{j \leq d}\{x \in [0,1]^{\infty} : x_j \in B_j\}$ for $d = 1, 2, \ldots$ and $B_j \in \mathcal{B}([0,1])$. Let $P_X$ be a probability measure defined on the measurable space $([0,1]^{\infty}, \mathcal{B}([0,1]^{\infty}))$ that is absolutely continuous to $\lambda^{\infty}$ and its Radon-Nikodym derivative satisfies $\|\frac{dP_X}{d\lambda^{\infty}}\|_{L^{\infty}([0,1]^{\infty})} < \infty$[1]. Then, suppose that there exists a true function $f^{\text{o}} : [0,1]^{\infty} \to \mathbb{R}$, and consider the following nonparametric regression problem with an infinite dimensional input:

$$Y = f^{\text{o}}(X) + \xi, \tag{1}$$

where $X$ is a random variable taking its value on $[0,1]^{\infty}$ and obeys the distribution $P_X$ introduced above, and $\xi$ is a observation noise generated from $N(0, \sigma^2)$ (a normal distribution with mean 0 and variance $\sigma^2 > 0$). Let $P$ be the joint distribution of $X$ and $Y$ obeying the regression model.

What we investigate in the following is (i) how efficiently we can approximate the true function $f^{\text{o}}$ by a neural network, and (ii) how accurately deep learning can estimate the true function $f^{\text{o}}$ from $n$ observations $D_n = (X_i, y_i)_{i=1}^n$ where $(X_i, y_i)_{i=1}^n$ are i.i.d. observations from the model. As a performance measure, we employ the mean squared error $\|f - f^{\text{o}}\|_{P_X}^2 := \mathrm{E}_P[(f(X) - f^{\text{o}}(X))^2]$, which can be seen as the excess risk of the predictive error $\mathrm{E}_{(X,Y) \sim P}[(f(X) - Y)^2]$ associated with the squared loss (i.e., $\|f - f^{\text{o}}\|_{P_X}^2 = \mathrm{E}_{(X,Y) \sim P}[(f(X) - Y)^2] - \mathrm{E}_{(X,Y) \sim P}[(f^{\text{o}}(X) - Y)^2] = \mathrm{E}_{(X,Y) \sim P}[(f(X) - Y)^2] - \inf_{f:\text{measurable}} \mathrm{E}_{(X,Y) \sim P}[(f(X) - Y)^2]$).

## 2.2 MIXED AND ANISOTROPIC SMOOTHNESS ON INFINITE DIMENSIONAL VARIABLES

Here, we introduce a function class in which we suppose the true function $f^{\text{o}}$ is included. For a given $l \in \mathbb{Z}_0^{\infty}$, define $\psi_{l_i} : [0,1] \to \mathbb{R}$ as $\psi_{l_i}(x) = \begin{cases} \sqrt{2}\cos(2\pi|l_i|x) & (l_i < 0), \\ \sqrt{2}\sin(2\pi|l_i|x) & (l_i > 0), \\ 1 & (l_i = 0), \end{cases}$ for $x \in [0,1]$, and define $\psi_l(X) := \prod_{i=1}^{\infty} \psi_{l_i}(x_i)$ for $X = (x_i)_{i=1}^{\infty} \in [0,1]^{\infty}$. Let $L^2([0,1]^{\infty}) := \{f : [0,1]^{\infty} \to \mathbb{R} : \int_{[0,1]^{\infty}} f^2(x)d\lambda^{\infty}(x) < \infty\}$ equipped with the inner product $\langle f, g \rangle := \int_{[0,1]^{\infty}} f(x)g(x)d\lambda^{\infty}(x)$ for $f, g \in L^2([0,1]^{\infty})$. Then, $(\psi_l)_{l \in \mathbb{Z}_0^{\infty}}$ forms a complete orthonormal system of $L^2([0,1]^{\infty})$, that is, $f \in L^2([0,1]^{\infty})$ can be expanded as $f(X) = \sum_{l \in \mathbb{Z}_0^{\infty}} \langle f, \psi_l \rangle \psi_l(X)$ (see Ingster & Stepanova (2011) for example). For $s \in \mathbb{N}_0^{\infty}$, let $\delta_s(f) : \mathbb{R}^{\infty} \to \mathbb{R}$ be

$$\delta_s(f)(\cdot) = \sum_{l \in \mathbb{Z}_0^{\infty} : \lfloor 2^{s_i-1} \rfloor \leq |l_i| < 2^{s_i}} \langle f, \psi_l \rangle \psi_l(\cdot),$$

---

[1]This is a rather strong assumption. We can omit this if we take $\theta = 1$ and $p = \infty$ for $\mathcal{F}_{p,\theta}^{\gamma}$. However, we don't pursue this direction in this study.

which can be seen as the frequency component of $f$ of frequency $|l_i| \simeq 2^{s_i}$ toward each coordinate. We also define $\|f\|_p := \left( \int_{[0,1]^\infty} |f|^p \mathrm{d}\lambda^\infty \right)^{1/p}$ for $p \geq 1$. Then, we define a function space with a general smoothness configuration as follows.

**Definition 1** (Function class with $\gamma$-smoothness). *For a given $\gamma : \mathbb{N}_0^\infty \to \mathbb{R}_{>0}$ which is monotonically non-decreasing with respect to each coordinate. For $p \geq 1$, $\theta \geq 1$, we define the $\gamma$-smooth space as*

$$\mathcal{F}_{p,\theta}^\gamma([0,1]^\infty) := \left\{ f = \sum_{l \in \mathbb{Z}_0^\infty} \langle f, \psi_l \rangle \psi_l : \left( \sum_{s \in \mathbb{N}_0^\infty} 2^{\theta\gamma(s)} \|\delta_s(f)\|_p^\theta \right)^{1/\theta} < \infty \right\},$$

*equipped with the norm $\|f\|_{\mathcal{F}_{p,\theta}^\gamma} := \left( \sum_{s \in \mathbb{N}_0^\infty} 2^{\theta\gamma(s)} \|\delta_s(f)\|_p^\theta \right)^{1/\theta}$.*

In the following, $\mathcal{F}_{p,\theta}^\gamma([0,1]^\infty)$ is abbreviated to $\mathcal{F}_{p,\theta}^\gamma$, and its unit ball is denoted by $U(\mathcal{F}_{p,\theta}^\gamma)$. Remind that $\delta_s(f)$ represents the frequency component associated with the frequency $(2^{s_i})_{i=1}^\infty$, and then the norm of the $\gamma$-smooth space imposes weight $2^{\theta\gamma(s)}$ on each frequency component associated with $s$. In that sense, $\gamma(s)$ controls the weight of each frequency component and accordingly a function in the space can have different smoothness toward different coordinates. As a special case of $\gamma(s)$, we investigate the following ones in this paper. We can see that a finite dimensional analysis can be easily reduced to a special case of the infinite dimensional analysis (see Appendix A). In that sense, our analysis generalizes existing finite dimensional analyses.

**Definition 2** (Mixed smoothness and anisotropic smoothness). *Given a monotonically non-decreasing sequence $a = (a_i)_{i=1}^\infty \in \mathbb{R}_{>0}^\infty$, we define the mixed smoothness as*

**(mixed smoothness)** $\qquad\qquad \gamma(s) = \langle a, s \rangle,$

*where $\langle a, s \rangle := \sum_{i=1}^\infty a_i s_i^2$, and define the anisotropic smoothness as*

**(anisotropic smoothness)** $\qquad \gamma(s) = \max\{a_i s_i : i \in \mathbb{N}\}.$

Each component $a_i$ of $a = (a_i)_{i=1}^\infty$ represents the smoothness of the function with respect to the variable $x_i$. Since we assumed $(a_i)_{i=1}^\infty$ is monotonically non-decreasing, a function in the space has higher smoothness toward the coordinate $x_i$ with higher index $i$. In other words, the function $f$ in the space is less sensitive to the variable $x_i$ with a larger index $i$. For example, in computer vision tasks, we may suppose $x_i$ with a large index $i$ corresponds to a higher frequency component of the input image, and then the function is less sensitive to such high frequency components and more sensitive to a low-frequency "global" information. This can be seen as an infinite dimensional variant of the mixed smooth Besov space (Schmeisser, 1987; Sickel & Ullrich, 2009) and the anisotropic Besov space (Nikol'skii, 1975; Vybiral, 2006; Triebel, 2011) (see Appendix C for detailed discussions). In our theoretical analysis, we will assume that the true target function $f^\circ$ is included in the $\gamma$-smooth function space.

**Assumption 3.** *The target function satisfies $f^\circ \in U(\mathcal{F}_{p,\theta}^\gamma)$ with $p \geq 1$ and $\theta \geq 1$, and $\|f^\circ\|_\infty \leq B_f$ for a fixed constant $B_f > 0$, where the smoothness $\gamma$ is either the mixed smoothness or the anisotropic smoothness.*

## 3 RELATION TO EXISTING WORK

A function space with the mixed smoothness in a finite dimensional setting can be found in Schmeisser (1987); Sickel & Ullrich (2009), in which the mixed smooth Besov space is defined. The approximation and estimation errors of deep neural networks for the mixed smooth Besov space were analyzed by Suzuki (2019) for a special setting of $a_1 = \cdots = a_d$, and an approximation error analysis for $a_1 = \cdots = a_d = 2$ was given by Montanelli & Du (2019) using a *sparse-grid* technique. The mathematical properties of the anisotropic Besov space with finite dimensional input were analyzed in Nikol'skii (1975); Vybiral (2006); Triebel (2011). The statistical analysis on the anisotropic Besov space can be dated back to Ibragimov & Khas'minskii (1984) and they derived the minimax

---

[2] Note that, since the number of nonzero components of $s \in \mathbb{N}_0^\infty$ is finite, the summation always converges. For the same reason, the maximum in the anisotropic smoothness is also attained by some finite index $i$.

optimal rate for density estimation where the density is in an anisotropic Besov space. Nyssbaum (1983; 1987) also analyzed a nonparametric regression problem on an anisotropic Besov space. The approximation and estimation error bounds by deep neural networks for composition functions in anisotropic Besove spaces and superiority of deep learning compared to the kernel methods are shown by Suzuki & Nitanda (2021). However, all of these studies are about finite dimensional input and it is far from trivial to generalize it to the infinite dimensional setting.

Our analysis for the $\gamma$-smooth function space is closely related to Ingster & Stepanova (2011) in which the anisotropic Sobolev space defined by $\mathcal{F}_c = \mathcal{W}_2^a := \left\{ f \in L^2([0,1]^\infty) : \sum_{i=1}^\infty \left\| \frac{\partial^{a_i} f}{\partial x_i^{a_i}} \right\|_2^2 < \infty \right\}$ is analyzed. They also derived a similar convergence rate to ours for non-deep learning estimator for a Gaussian white noise model. In the literature of the *functional data analysis*, the Nadaraya-Watson estimator for functional input has been extensively studied (Ferraty et al. (2007) and Ling & Vieu (2018) for a comprehensive survey). If we apply the bound given in the literature to our setting, the learning rate can be $1/\text{poly-}\log(n)$ which is much slower than our analysis. This is because their analysis does not make use of $\gamma$-smoothness. See Appendix C for more details.

Kohler & Langer (2020) analyzed CNNs in a setting where the target function has a hierarchical max-pooling structure each layer of which is sufficiently smooth. On the other hand, our $\gamma$-smooth function class imposes smoothness more directly on the target function. Liu et al. (2021) analyzed learning ability of CNNs with a ResNet structure in a classification task where the data are distributed on a low-dimensional manifold and established a rate which only depends on the dimensionality of the low dimensional manifold. However, the input should be distributed on a low dimensional manifold, while our analysis allows its support to be infinite dimensional. Estimation errors on a low dimensional structure also have been studied in Yang & Dunson (2016); Bickel & Li (2007); Nakada & Imaizumi (2020); Schmidt-Hieber (2019); Chen et al. (2019b).

## 4 DEFINITION OF A DILATED CONVOLUTIONAL NEURAL NETWORK

In this section, we introduce the neural network model that we investigate in this paper. Let $L \in \mathbb{N}$ be the depth of the network and $d_i$ ($i = 1, \ldots, L+1$) be the width of the $i$-th layer in the network where we set $d_{L+1} = 1$. Then, the fully connected neural network (FNN) can be given by $(A_L \eta(\cdot) + b_L) \circ \cdots \circ (A_i \eta(\cdot) + b_i) \circ \cdots \circ (A_1 x + b_1)$ where $A_i \in \mathbb{R}^{d_{i+1} \times d_i}$, $b_i \in \mathbb{R}^{d_{i+1}}$ and $\eta(x) = \max\{x, 0\}$ is the ReLU activation function that is applied element-wise. The set of FNN with depth $L \in \mathbb{N}$, maximum width $W \in \mathbb{N}$, norm bound $B > 0$, and sparsity level $S \in \mathbb{N}$ is defined by

$$\Phi(L, W, S, B) := \Big\{ f(x) = (A_L \eta(\cdot) + b_L) \circ \cdots \circ (A_i \eta(\cdot) + b_i) \circ \cdots \circ (A_1 x + b_1) :$$

$$\max_{i=1,\ldots,L} \|A_i\|_\infty \vee \|b_i\|_\infty \leq B, \ \sum_{i=1}^L \|A_i\|_0 + \|b_i\|_0 \leq S, \ \max_{i=1,\ldots L} d_i \leq W \Big\},$$

where $\| \cdot \|_\infty$ is the maximum absolute value among the elements of a vector or matrix[3], and $\| \cdot \|_0$ is the number of non-zero elements of a vector or matrix.

Next, we define the (dilated) CNNs. Let $C \in \mathbb{N}$ be the number of channels and $\mathbb{R}^{C \times \infty} := \left\{ (x_1, \ldots x_i, \ldots) : x_i \in \mathbb{R}^C \right\}$. Suppose that $w \in \mathbb{R}^{C \times W'}$ is a filter with a width $W' \in \mathbb{N}$, channel size $C \in \mathbb{N}$ and an interval $h \in \mathbb{N}$, then define the *dilated convolution* $w \star_h X' \in \mathbb{R}^\infty$ for an infinite-sequence of vectors $X' = (x'_{i,j})_{i=1,j=1}^{C,\infty} \in \mathbb{R}^{C \times \infty}$ as $(w \star_h X')_k = \sum_{i=1}^C \sum_{j=1}^{W'} w_{i,j} x'_{i,h(j-1)+k}$. When $h = 1$, it is called a normal convolution. Moreover, given a filter $F \in \mathbb{R}^{C' \times C \times W'}$ with $(C')$-multiple channel outputs, we define its corresponding convlution $\text{Conv}_{h,F} : \mathbb{R}^{C \times \infty} \to \mathbb{R}^{C' \times \infty}$ as

$$\text{Conv}_{h,F}(X') = \begin{pmatrix} F_{1,:,:} \star_h X' \\ \vdots \\ F_{C',:,:} \star_h X' \end{pmatrix}.$$

Then, the *dilated CNN* can be defined as follows (its illustration can be found in Figure 1).

---

[3]We define $a \vee b := \max\{a, b\}$ and $a \wedge b := \min\{a, b\}$ for $a, b \in \mathbb{R}$.

**Definition 4** (Dilated CNN). *For a given $L'$, $W' \in \mathbb{N}$, suppose that we are given filters $F_l \in \mathbb{R}^{C_{l+1} \times C_l \times W'}$ with the number of channels $C_l \in \mathbb{N}$ ($l \in [L']$) with $C_1 = 1$ and an FNN $g_{\mathrm{FNN}} \in \Phi(L, W, B, S)$, then a neural network given by $f(X) = \left( g_{\mathrm{FNN}} \circ \mathrm{Conv}_{W'^{L'-1}, F_{L'}} \circ \cdots \circ \mathrm{Conv}_{W'^{l-1}, F_l} \circ \cdots \circ \mathrm{Conv}_{1, F_1} \circ X \right)_1$ is called a* dilated CNN[4], *where $g_{\mathrm{FNN}}$ is assumed to be applied in an element-wise manner to the infinite sequence. The set of dilated CNNs with the same number of channels $C_l = C'$ ($2 \leq \forall l \leq L'$) in all layers but $C_1 = 1$ is denoted by*

$$\mathcal{P}(L', B', W', C', L, W, S, B) = \left\{ \left( g_{\mathrm{FNN}} \circ \mathrm{Conv}_{W'^{L'-1}, F_{L'}} \circ \cdots \circ \mathrm{Conv}_{1, F_1} \circ X \right)_1 : \right.$$
$$\left. F_l \in \mathbb{R}^{C' \times C' \times W'} \ (l \geq 2), \ F_1 \in \mathbb{R}^{C' \times 1 \times W'}, \ \|F_l\|_\infty \leq B', \ g_{\mathrm{FNN}} \in \Phi(L, W, B, S) \right\}.$$

For simplicity, the set of dilated CNNs is abbreviated to $\mathcal{P}$ when there is no ambiguity about the parameter configuration. When $L' = 1$, it coincides with a set of regular CNNs. In our analysis, it is sufficient to consider an dilated CNN with a constant number of channels throughout all layers ($C_l = C$ ($\forall l \in [L']$)). To evaluate the estimation accuracy, it is important to assume the functions in the set is bounded in terms of the $L_\infty$-norm. For that purpose, we consider an dilated CNN clipped by a bound $B_f > 0$ defined as $\bar{\mathcal{P}}(B_f, L', B', W', C, L, W, S, B) := \left\{ \bar{f}(X) = (-B_f \vee (B_f \wedge f(X))) : f \in \mathcal{P}(L', B', W', C, L, W, S, B) \right\}$.

**Remark 5.** *In the definition of the dilated CNN, we do not impose ReLU activation. However, since ReLU activation can realize a linear function for a bounded input and thus our analysis can be straightforwardly applied even if there is nonlinear ReLU activation. Moreover, this paper mainly focuses on 1D-convolution, but it can be generalized to* **2D-convolution**. *See Appendix I for the detailed discussions in which it is shown that $\gamma$-smoothness over a wavelet decomposition of an input image achieves the same rate of convergence as in 1D-convolution.*

## 5 Approximation and estimation errors of deep learning

In this section, we give our main result about the approximation and estimation errors of FNNs and dilated CNNs when the true function $f^\circ$ is in the $\gamma$-smooth function class.

### 5.1 Approximation error analysis by fully connected neural networks

Here, we present the approximation error analysis of FNNs for a general smoothness $\gamma$ not restricted to the mixed/anisotropic smoothness. For a given $T > 0$ and the smoothness $\gamma : \mathbb{N}_0^\infty \to \mathbb{R}_{>0}$, define

$$I(T, \gamma) := \left\{ i \in \mathbb{N} : \exists s \in \mathbb{N}_0^\infty, \ s_i \neq 0, \ \gamma(s) < T \right\},$$

and then the following quantities play an important role in our approximation error analysis.

**Definition 6** (Axial complexity and frequency direction complexity). *The* axial complexity *is defined by $d_{\max}(T, \gamma) := |I(T, \gamma)|$. Moreover, the* frequency direction complexity *is defined by $f_{\max}(T, \gamma) := \max_{s \in \mathbb{N}_0^\infty : \gamma(s) \leq T} \max_{i \in \mathbb{N}} s_i$.*

The axial complexity is used to evaluate how many components need to be extracted from a given infinite-dimensional sequence $X \in \mathbb{R}^\infty$ to achieve a particular approximation error, and the frequency complexity characterizes up to which frequency we require to approximate a target function with a particular error. Let

$$v := \left( \frac{1}{p} - \frac{1}{2} \right)_+, \ \alpha(\gamma) := \sup_{s \in \mathbb{N}_0^\infty} \frac{\sum_{i=1}^\infty s_i}{\gamma(s)}, \ G(T, \gamma) := \sum_{s \in \mathbb{N}_0^\infty : \gamma(s) < T} 2^s,$$

where $(x)_+ := \max\{x, 0\}$. Then, a general approximation error theory for FNNs can be obtained as follows.

**Theorem 7** (Approximation error for the $\gamma$-smooth space by FNNs). *Assume that $\gamma$, $\gamma' : \mathbb{N}_0^\infty \to \mathbb{R}_{>0}$ satisfy*

$$\gamma'(s) < \gamma(s), \ v\alpha(\gamma) < 1, \ v\alpha(\gamma') < 1,$$

---

[4]Here, we employ $h = W'^{k-1}$ for the $k$-th layer convolution. This structure is useful to show its feature extraction ability in Section 5.3.

and the target function $f \in \mathcal{F}_{p,\theta}^{\gamma}$ $(p \geq 1, \theta \geq 1)$ to be approximated satisfies $\|f\|_{\infty} \leq B_f$ for a constant $B_f \in \mathbb{R}_{>0}$. For arbitrary $T > 0$, we let a tuple $(d_{\max}, f_{\max}, G)$ be

$$(d_{\max}, f_{\max}, G) = \begin{cases} (d_{\max}(\gamma), f_{\max}(\gamma), G(T, \gamma)) & (1 \leq \theta \leq 2), \\ (d_{\max}(\gamma'), f_{\max}(\gamma'), G(T, \gamma')) & (2 < \theta), \end{cases}$$

and with some positive constants $K$, $K'$ depending only on $B_f$, we let

$$L = 2K \max \left\{ d_{\max}^2, T^2, (\log G)^2, \log f_{\max} \right\}, \qquad W = 21 d_{\max} G,$$

$$S = 1764 K d_{\max}^2 \max \left\{ d_{\max}^2, T^2, (\log G)^2, \log f_{\max} \right\} G, \qquad B = (\sqrt{2})^{d_{\max}} K'.$$

Then, there exists an FNN $\hat{R}_T \in \Phi(L, W, S, B)$ with $d_{\max}$-dimensional input that takes $(x_i)_{i \in I(T, \gamma)} \in [0, 1]^{d_{\max}}$ as an input such that $f' : [0, 1]^{\infty} \to \mathbb{R}$ given by $f'(X) := \hat{R}_T \left( (x_i)_{i \in I(T, \gamma)} \right)$ for $X = (x_i)_{i=1}^{\infty} \in [0, 1]^{\infty}$ satisfies

$$\|f - f'\|_2 \lesssim \begin{cases} 2^{-(1 - v\alpha(\gamma))T} \|f\|_{\mathcal{F}_{p,\theta}^{\gamma}} & (1 \leq \theta \leq 2), \\ 2^{-(1 - v\alpha(\gamma'))T} \left( \sum_{T \leq \gamma'(s)} 2^{\frac{2\theta}{\theta - 2}(\gamma'(s) - \gamma(s))} \right)^{1/2 - 1/\theta} \|f\|_{\mathcal{F}_{p,\theta}^{\gamma}} & (2 < \theta). \end{cases}$$

According to this theorem, the derived approximation error can be achieved by FNNs if the required $d_{\max}$ components of the input $X$ is extracted. This theorem clarifies how the decay rate of the frequency components of the target function affects the approximation accuracy. Since the approximation accuracy is determined by $(d_{\max}, f_{\max}, G)$, it is not directly affected by the dimensionality but is characterized merely by the smoothness parameter $\gamma$. Intuitively, $T > 0$ controls the approximation accuracy and simultaneously controls up to which frequency is used for the approximation. Specifically, the difficulty of the approximation is determined by the number of bases required that is characterized by the number of $s \in \mathbb{N}_0^{\infty}$ with $\gamma(s) < T$, and the maximum frequency required for the approximation is also important for the analysis. The bound is proven by evaluating an approximation error of a trigonometric polynomial approximation of $f \in \mathcal{F}_{p,\theta}^{\gamma}$ and showing that we can construct a neural network that approximates a trigonometric polynomial with a certain accuracy.

## 5.2 SMOOTHNESS WITH POLYNOMIAL ORDER INCREASE

Here, we derive a concrete convergence rate for CNNs in a setting where $\gamma$ is mixed or anisotropic smoothness and the smoothness parameter $a = (a_i)_{i=1}^{\infty}$ is polynomially increasing. In this setting, we just need to use only one layer CNN. Deeper CNN layers will be used in the next section (Section 5.3) to perform adaptive feature extraction from wide range of inputs.

**Assumption 8.** *There exists $0 < q < \infty$ such that the smoothness parameter $a = (a_i)_{i=1}^{\infty}$ satisfies $a_i = \Omega(i^q)$. We also assume $a_1 < a_2$ for the mixed smoothness setting.*

This assumption impose that the target function should be sufficiently smooth with respect to higher order indices. Under this setting, we show the approximation and estimation errors as follows. First, the approximation error by the CNNs can be evaluated as follows.

**Theorem 9** (Approximation error bound under smoothness with polynomial order increase)**.** *Suppose that Assumptions 3 and 8 hold, then we have the following approximation error bounds:*

1. Mixed smoothness $(\gamma(s) = \langle a, s \rangle)$: *Suppose that $v/a_1 < 1$. Then, for arbitrary $T > 0$, there exists a configuration of the network structure, $L' = 1$, $B' = 1$, $W' \sim T^{\frac{1}{q}}$, $C' \sim T^{\frac{1}{q}}$ and*

$$L_1(T) \sim \max \left\{ T^{\frac{2}{q}}, T^2 \right\}, \quad W_1(T) \sim \left( \prod_{i=2}^{\infty} \left( 1 - 2^{\frac{-(a_i - a_1)}{a_1}} \right)^{-1} \right) T^{\frac{1}{q}} 2^{\frac{T}{a_1}},$$

$$S_1(T) \sim \left( \prod_{i=2}^{\infty} \left( 1 - 2^{\frac{-(a_i - a_1)}{a_1}} \right)^{-1} \right) T^{\frac{2}{q}} \max \left\{ T^{\frac{2}{q}}, T^2 \right\} 2^{\frac{T}{a_1}}, \quad B_1(T) \sim (\sqrt{2})^{T^{\frac{1}{q}}},$$

*such that there exists an dilated CNN $f' \in \mathcal{P}(L', B', W', C', L_1(T), W_1(T), S_1(T), B_1(T))$ satisfying the following approximation error:*

$$\|f' - f^{\circ}\|_2 \lesssim 2^{-\left(1 - \frac{v}{a_1}\right)T}.$$

2. Anisotropic smoothness $(\gamma(s) = \max_i \{a_i s_i\})$: *Let $\tilde{a} := (\sum_{i=1}^{\infty} a_i^{-1})^{-1}$ and suppose $0 < \tilde{a}$ and $v < \tilde{a}$, then there exists a network structure setting $L' = 1$, $B' = 1$, $W' \sim T^{\frac{1}{q}}$, $C' \sim T^{\frac{1}{q}}$ and*

$$L_2(T) \sim \max \left\{ T^{\frac{2}{q}}, T^2 \right\}, \quad W_2(T) \sim T^{\frac{1}{q}} 2^{T/\tilde{a}}, \quad S_2(T) \sim T^{\frac{2}{q}} \max \left\{ T^{\frac{2}{q}}, T^2 \right\} 2^{T/\tilde{a}}, \quad B_2(T) \sim (\sqrt{2})^{T^{\frac{1}{q}}},$$

*such that there exists an dilated CNN $f' \in \mathcal{P}(L', B', W', C', L_2(T), W_2(T), S_2(T), B_2(T))$ satisfying the following approximation error: $\|f' - f^\circ\|_2 \lesssim 2^{-(1-v/\tilde{a})T}$.*

The proof can be found in Appendix E. From this theorem, we can see that the number of layers, the width, the number of parameters, and the size of the parameters are both determined by $T$ and the smoothness parameter $a$. Moreover, in Theorem 7, the approximation error was derived assuming that the appropriate index set $I(T, \gamma)$ was provided. On the other hand, in Theorem 9, we do not make such an assumption because the CNNs can automatically extract the required index $I(T, \gamma)$.

Next, we consider the estimation error of these models in the regression problem (Eq. (1)). Suppose that we are given $n$ observations $D_n = (X_i, y_i)_{i=1}^n$ following the model (1). We consider the empirical risk minimization estimator (ERM estimator) in the model $\mathcal{P}$ that is given by any minimizer of the empirical risk:

$$\hat{f} \in \underset{f \in \bar{\mathcal{P}}}{\operatorname{argmin}} \frac{1}{n} \sum_{i=1}^n (f(X_i) - y_i)^2.$$

As we have stated above, we employ the mean squared error $\|\hat{f} - f^\circ\|_{P_X}^2$ as a performance measure. Since $\hat{f}$ depends on the training data $D_n$, we take expectation with respect to $D_n$: $\mathrm{E}_{P^n}[\|\hat{f} - f^\circ\|_{P_X}^2] := \mathrm{E}_{(X_i, y_i)_{i=1}^n \sim P^n}[\|\hat{f} - f^\circ\|_{P_X}^2]$. Then, the following theorem holds.

**Theorem 10** (Estimation error under smoothness with polynomial order increase). *Suppose that Assumptions 3 and 8 hold, then we have the following estimation error bounds:*

1. Mixed smoothness ($\gamma(s) = \langle a, s \rangle$): *If $v/a_1 < 1$, then by setting the network structure as $L' = 1$, $B' = 1$, $W' \sim (\log n)^{\frac{1}{q}}$, $C' \sim (\log n)^{\frac{1}{q}}$ and $(L, W, S, B) = (L_1(T), W_1(T), S_1(T), B_1(T))$ for $T = \frac{a_1}{2(a_1 - v) + 1} \log_2(n)$, the ERM estimator $\hat{f}$ in $\bar{\mathcal{P}}(B_f, L', B', W', C', L, W, S, B)$ achieves*

$$\mathrm{E}_{P^n}[\|\hat{f} - f^\circ\|_{P_X}^2] \lesssim \left( \prod_{i=2}^\infty (1 - 2^{-\frac{(a_i - a_1)}{a_1}})^{-1} \right) n^{-\frac{2(a_1 - v)}{2(a_1 - v) + 1}} (\log n)^{\frac{2}{q} + 2} \max\{(\log n)^{\frac{4}{q}}, (\log n)^4\}.$$

2. Anisotropic smoothness ($\gamma(s) = \max_i\{a_i s_i\}$): *Under the same setting, if $v < \tilde{a}$, by setting the network structure as $L' = 1$, $B' = 1$, $W' \sim (\log n)^{\frac{1}{q}}$, $C' \sim (\log n)^{\frac{1}{q}}$ and $(L, W, S, B) = (L_2(T), W_2(T), S_2(T), B_2(T))$ for $T = \frac{\tilde{a}}{2(\tilde{a} - v) + 1} \log_2(n)$, the ERM estimator $\hat{f}$ in $\bar{\mathcal{P}}(B_f, L', B', W', C', L, W, S, B)$ achieves*

$$\mathrm{E}_{P^n}[\|\hat{f} - f^\circ\|_{P_X}^2] \lesssim n^{-\frac{2(\tilde{a} - v)}{2(\tilde{a} - v) + 1}} (\log n)^{\frac{2}{q} + 2} \max\{(\log n)^{\frac{4}{q}}, (\log n)^4\}.$$

The proof can be found in Appendix F. This theorem shows that even if the dimension of the input data is infinite, for a function with a particular smoothness, CNNs can achieve a dimension-independent convergence rate which is a polynomial order, that is, it can avoid the curse of dimensionality by utilizing the increasing smoothness. We can see that the derived convergence rate is a direct extension of finite dimensional one. Actually, if $v = 0$, the rate for the anisotropic smoothness matches that of the finite dimensional one (Suzuki & Nitanda, 2021) up to poly-log order which is known as minimax optimal. Therefore, CNNs can achieve the optimal rate up to poly-log order at least when $v = 0$. As for the mixed smoothness, a finite dimensional version was analyzed (Suzuki, 2019) and a similar rate was derived. However, our analysis assumes $a_1 < a_2$ and $a_i = \Omega(i^q)$ and thus obtained completely dimensionality independent bound while the bound by Suzuki (2019) depends on $d$ in the exponent of the poly-log order.

## 5.3 SMOOTHNESS WITH SPARSITY

Next, we relax the assumption $a_i = \Omega(i^q)$ and consider a situation where there is a kind of sparse structure in $a$. As we have seen in the previous section, under the assumption that the coordinates with large indices are not important, polynomial-order convergence rate depending only on the smoothness can be achieved by the ordinary CNNs. In this section, we show that similar rates can be achieved by using dilated CNNs even when $a$ does not satisfy the polynomial order increase if $a$ has *sparsity*. For that purpose, we first define the sparsity of the smoothness.

**Definition 11** (Weak $\ell^q$-norm of smoothness). *Given $a = (a_i)_{i=1}^\infty \in \mathbb{R}_{>0}^\infty$ which is not necessarily monotonically increasing, consider the sorted sequence $0 < a_{i_1} \leq a_{i_2} \leq \cdots$ in the ascending order. Then, define its weak $\ell^q$-norm for $0 < q < \infty$ as $\|a\|_{wl^q} := \sup_j j^q a_{i_j}^{-1}$.*

This kind of sparsity inducing norm were introduced and discussed previously in Donoho (1993); Donoho et al. (1996); Yang & Barron (1999) to quantify sparsity of coefficients of basis expansions. We notice that, if $\|a\|_{wl^q}$ is small, almost all $a_i$s are very large and there are only few indices that are small, which means sparseness. If the smoothness parameter $a$ has a small weak $\ell^q$-norm, then we can say that the functions with such smoothness has a small number of important coordinate directions. Therefore, it is expected that we can approximate such a function efficiently by a neural network. In this section, we analyze the approximation and estimation errors under the condition of sparse smoothness.

**Assumption 12.** *$a = (a_i)_{i=1}^{\infty}$ satisfies $\|a\|_{wl^q} \leq 1$ for $0 < q < \infty$ and $a_i = \Omega(\log i)$.*

Note that this assumption relaxes the condition $a_i = \Omega(i^q)$ in Assumption 8 to $a_i = \Omega(\log i)$. Instead, it imposes the sparsity $\|a\|_{wl^q} \leq 1$. Under this assumption, we obtain the following approximation error bound.

**Theorem 13** (Approximation error bound for sparse smoothness). *Suppose that Assumptions 3 and 12 hold, then we have the following approximation error bounds for any $T > 1$:*

1. Mixed smoothness ($\gamma(s) = \langle a, s \rangle$): *Suppose that $v/a_{i_1} < 1$, then there exist a set of network structure parameters satisfying*[5]

$$L' \sim T,\ B' = 1,\ W' = 3,\ C' \sim T^{\frac{1}{q}},$$

*such that there exists an dilated CNN $f' \in \mathcal{P}(L', B', W', C', L_1(T), W_1(T), S_1(T), B_1(T))$ satisfying $\|f' - f^{\circ}\|_2 \lesssim 2^{-(1 - \frac{1}{a_{i_1}})T}$.*

2. Anisotropic smoothness ($\gamma(s) = \max_i \{a_i s_i\}$): *Suppose that $0 < \tilde{a}$ and $v < \tilde{a}$, then there exist a set of network structure parameters satisfying $L' \sim T,\ B' \sim 1,\ W' = 3,\ C' \sim T^{\frac{1}{q}}$ such that there exists an dilated CNN $f' \in \mathcal{P}(L', B', W', C', L_2(T), W_2(T), S_2(T), B_2(T))$ satisfying $\|f' - f^{\circ}\|_2 \lesssim 2^{-(1 - v/\tilde{a})T}$.*

We can see that this approximation error bound gives the same bound as Theorem 9 under a relaxed condition Assumption 12 with sparse smoothness. The only difference is the setting of the convolution part $(L', W', C)$ and other parts are same as Theorem 9. This difference is required to find the important indices that are relatively non-smooth compared with other indices. Thanks to the structure of dilated convolution, it can find such indices from a long range of index set: $\{i \in \mathbb{N} : i = O(3^{L'})\}$ (see Figure 1 for illustration of feature extraction by CNNs). Accordingly, we also have the following estimation error bound.

**Theorem 14** (Estimation error for sparse smoothness). *Suppose that Assumptions 3 and 12 hold, then by setting $L' \sim \log n,\ B' \sim 1,\ W' = 3,\ C' \sim (\log n)^{\frac{1}{q}}$ and $(L, W, S, B)$ as in Theorem 10, the ERM estimator $\hat{f}$ in the class of dilated CNNs can achieve the same convergence rate of the estimation error as Theorem 10.*

The proof can be found in Appendix G. These theorems show that for polynomially increasing smoothness, ordinary CNNs can perform optimal coordinate selection, while for sparse smoothness, dilated CNNs play an important role in coordinate selection. This theorem shows that, when extracting data with long-term dependence, convergence rates that avoid dependence on dimensionality can be achieved by using dilated CNNs.

## 6 CONCLUSION

In this study, we gave a condition on the smoothness of the function space as one of the situations in which the curse of dimensionality can be avoided when the input is ultra-high dimensional ($n \ll d$) or infinite dimensional ($d = \infty$). This study showed that the smoothness of the target function plays an essential role in characterizing the estimation error bound. Especially, when the smoothness parameter $(a_i)_{i=1}^{\infty}$ grows up as the index $i$ increase, we can obtain a polynomial order convergence rate even if the input is infinite dimensional. Future plans include, for example, considering a situation where the smoothness depends on each input location, and extending the definition of $\mathcal{F}_{p,\theta}^{\gamma}$ so that it captures more realistic situations.

---

[5]The choice $W' = 3$ is not a strict requirement. It can be replaced by an arbitrary integer $H$ with $H \geq e$.

ACKNOWLEDGMENT

This study was partially supported by JSPS KAKENHI (18H03201), Japan Digital Design and JST CREST.

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

———Appendix———

## NOTATION LISTS

Table 2: Notation list

| notation | definition |
|---|---|
| $n$ | sample size |
| $(X_i, y_i)$ | $i$-th observation ($X_i$: input, $y_i$: output) |
| $D_n = (X_i, y_i)_{i=1}^n$ | training data |
| $f^{\circ}$ | the true function |
| $a = (a_i)_{i=1}^{\infty}$ | smoothness with respect to each coordinate |
| $s = (s_i)_{i=1}^{\infty}$ | frequency with respect to each coordinate |
| $\psi_l(X) = \prod_{i=1}^{\infty} \psi_{l_i}(x_i)$ | trigonometric orthonormal basis functions |
| $\delta_s(f)$ | basis function expansion of $f$ for $l \in \mathbb{Z}_0^{\infty}$ such that $\lfloor 2^{s_i-1} \rfloor \leq |l_i| < 2^{s_i}$ |
| $\|\cdot\|_2$ | $L_2$-norm with respect to the uniform distribution ($\|f\|_2 := \sqrt{\int f(X)^2 \mathrm{d}\lambda^{\infty}(X)}$) |
| $\|\cdot\|_{P_X}$ | $L_2$-norm with respect to $P_X$ ($\|f\|_{P_X} := \sqrt{\mathrm{E}_{X \sim P_X}[f(X)^2]}$) |
| $\mathcal{F}_{p,\theta}^{\gamma}([0,1]^{\infty})$ | $\gamma$-smooth function class |
| $\gamma(s)$ | penalty on each frequency component |
| $\gamma(s) = \langle a, s \rangle$ | mixed smoothness |
| $\gamma(s) = \max\{a_i s_i : i \in \mathbb{N}\}$ | anisotropic smoothness |
| $\eta$ | ReLU activation function |
| $\Psi(L, W, S, B)$ | set of fully connected networks with depth $L$, width $W$, sparsity level $S$, norm bound $B$ |
| $\mathcal{P}(L', B', W', C', L, W, S, B)$ | set of dilated CNNs with depth $L'$, filter width $W'$, channel size $C'$ accompanied with an FNN in $\Psi(L, W, S, B)$ |
| $I(T, \gamma)$ | the set of features contributing a frequency component $s$ with $\gamma(s) < T$ |
| $d_{\max}(T, \gamma)$ | $|I(T, \gamma)|$: axial complexity |
| $f_{\max}(T, \gamma)$ | $\max_{s \in \mathbb{N}_0^{\infty}: \gamma(s) \leq T} \max_{i \in \mathbb{N}} s_i$: frequency direction complexity |
| $v$ | $\left(\frac{1}{p} - \frac{1}{2}\right)_+$ |
| $\alpha(\gamma)$ | $\sup_{s \in \mathbb{N}_0^{\infty}} \frac{\sum_{i=1}^{\infty} s_i}{\gamma(s)}$ |
| $G(T, \gamma)$ | $\sum_{s \in \mathbb{N}_0^{\infty}: \gamma(s) < T} 2^s$ |

## A  CONNECTION TO FINITE DIMENSIONAL INPUT SETTING

We can easily see that the analysis in our paper can be directly applied to a setting where the input is finite dimensional (say, $d$-dimensional). Let

$$J_d := \left\{ s \in \mathbb{N}_0^{\infty} : s_i = 0 \ (i = d+1, \dots) \right\},$$

and suppose that $\gamma(s) : \mathbb{N}^{\infty} \to \mathbb{R}_{>0} \cup \{\infty\}$ satisfies $\gamma(s) < \infty$ ($s \in J_d$) and $\gamma(s) = \infty$ ($s \notin J_d$). If we set

$$\mathcal{F}_{p,\theta,d}^{\gamma}([0,1]^{\infty}) := \left\{ f = \sum_{l \in \mathbb{Z}_0^{\infty}} \langle f, \psi_l \rangle \psi_l : \left( \sum_{s \in J_d} 2^{\theta\gamma(s)} \|\delta_s(f)\|_p^{\theta} \right)^{1/\theta} < \infty, \ \delta_s(f) = 0 \ (\forall s \notin J_d) \right\},$$

then by the condition $\delta_s(f) = 0$ ($\forall s \notin J_d$), $f \in \mathcal{F}_{p,\theta,d}^{\gamma}([0,1]^{\infty})$ is independent of $(x_i)_{i>d}$:

$$f(x_1, \dots, x_d, x_{d+1}, \dots, x_i, \dots) = f(x_1, \dots, x_d, x'_{d+1}, \dots, x'_i \dots),$$

for any $(x_i)_{i=1}^{\infty} \in [0,1]^{\infty}$ and $x'_i \in [0,1]$ ($i = d+1, d+2, \dots$). Then, for $f \in \mathcal{F}_{p,\theta,d}^{\gamma}([0,1]^{\infty})$, we may define

$$f_d(x_1, \dots x_d) := f(x_1, \dots, x_d, 0, \dots),$$

and it holds that $\|f - f'\|_2 = \|f_d - f'_d\|_2$. Hence, by the same argument as the proof of Theorem 7, we have the same bound for $\mathcal{F}^\gamma_{p,\theta,d}([0,1]^\infty)$. Thus, the same statement as Theorem 7 holds for the $\gamma$-smooth functions on $[0,1]^d$. This shows that the difficulty of the approximation depends only on the smoothness $\gamma(s)$ and independent of the dimensionality $d$. In particular, we can establish the bound even for $d \gg n$.

# B  ANALYSIS OF ESTIMATION ERROR BY CONVOLUTIONAL NEURAL NETWORK

In this section, we discuss approximation and estimation errors by CNNs and dilated CNNs. In Theorem 7, we considered approximating a function in $\mathcal{F}^\gamma_{p,\theta}$ by FNNs. According to the analysis, the index required to achieve the derived approximation error bound is determined by $I(T, \gamma)$, and only the coordinates corresponding to the index set $I(T, \gamma)$ in $X$ should be taken as input. However, in practice, it is not given which index is required as input, and it should be estimated from the data. In this section, we show that, under certain conditions, it is possible to find the required indices from the data by using CNN and dilated CNN type architectures. We also show that these architectures can achieve a favorable approximation and estimation errors for functions in mixed and anisotropic smooth spaces that depends on the smoothness of the function classes.

# C  RELATIONSHIP TO EXISTING FUNCTION SPACES

In this section, we discuss the relationship between the space $\mathcal{F}^\gamma_{p,\theta}$, its finite dimensional counter-part and some related function classes.

## C.1  MIXED SMOOTHNESS

First, we introduce a finite dimensional counter part of or function space with mixed smoothness (Schmeisser, 1987; Sickel & Ullrich, 2009), where the domain of the input is $[0,1]^d$.

**Definition 15** (Mixed smooth modulus of smoothness). *For $r \in \mathbb{N}$ and $h \in \mathbb{R}_{>0}$, let*

$$\Delta_h^r(f)(x) = \begin{cases} \sum_{j=0}^r \binom{r}{j}(-1)^{r-j} f(x+jh) & (x \in [0,1],\ x+rh \in [0,1]), \\ 0 & (otherwise), \end{cases}$$

*be the $r$-th order discrete differentiation for a function $f : [0,1] \to \mathbb{R}$. By applying this discrete differential operator to each coordinate of a subset $e \subset \{1, \ldots, d\}$, the mixed discrete differential operator for a step length $h \in \mathbb{R}^d_{>0}$ and the order parameter $r \in \mathbb{N}^d$ is defined as*

$$\Delta_{h_i}^{r_i,i}(f)(x) = \Delta_{h_i}^{r_i}\left(f(x_1, \ldots, \cdot, \ldots, x_d)(x_i))\right),\ \Delta_h^{r,e}(f) := \left(\prod_{i \in e} \Delta_{h_i}^{r_i,i}\right)(f),$$

*for $x \in [0,1]^d$. Then, the modulus of mixed smoothness is defined by*

$$w_{r,p}^e(f,t) := \sup_{|h_i| \leq t_i, i \in e} \|\Delta_h^{r,e}(f)\|_p,$$

*for $t \in \mathbb{R}^d_{>0}$ and $1 \leq p$, where $\|\cdot\|_p$ is the $L_p$-norm with respect to the Lebesgue measure on $[0,1]^d$.*

Then, based on this modulus of mixed smoothness, we can define the mixed smooth Besov space as follows.

**Definition 16** (Mixed smooth Besov space). *Let $1 \leq p, q \leq \infty$. For a given smoothness parameter $a \in \mathbb{R}^d_{>0}$, let $r_i = \lfloor a_i \rfloor + 1$ $(i \in [d])$ and define the seminorm $|\cdot|_{MB^{a,e}_{p,q}}$ as*

$$|f|_{MB^{a,e}_{p,q}} := \begin{cases} \left\{\int_{x \in [0,1]^d} \left[\left(\prod_{i \in e} t_i^{-a_i}\right) w_{r,p}^e(f,t)\right]^q \frac{dt}{\prod_{i \in e} t_i}\right\}^{1/q} & (1 \leq q < \infty), \\ \sup_{t \in [0,1]^d} \left(\prod_{i \in e} t_i^{-a_i}\right) w_{r,p}^e(f,t) & (q = \infty). \end{cases}$$

*Then we define the norm of the mixed smooth Besov space as*

$$\|f\|_{MB^a_{p,q}} := \|f\|_p + \sum_{e \subset \{1, \ldots, d\}} |f|_{MB^{a,e}_{p,q}},$$

so that the mixed smooth Besov space is given by $MB_{p,q}^{\alpha}([0,1]^d) := \{f \in L^p([0,1]^d) : \|f\|_{MB_{p,q}^a} < \infty\}$.

The mixed smooth Besov space defined above can be seen as the finite dimensional counter part of our space with mixed smoothness $\gamma$. Here, for $s \in \mathbb{N}^d$, let

$$\delta_s(f) = \sum_{l \in \mathbb{Z}_0^d : \lfloor 2^{s_i-1} \rfloor \le |l_i| < 2^{s_i}} \langle f, \psi_l \rangle \psi_l,$$

where $\psi_l : [0,1]^d \to \mathbb{R}$ is defined in the same way as in the infinite dimensional setting and $f \in L^2([0,1]^d)$. Then, it is known that

$$|f|_{MB_{p,q}^a} \sim \left( \sum_{s \in \mathbb{N}_0^d} (2^{\langle a,s \rangle} \|\delta_s(f)\|_p)^q \right)^{1/q}$$

holds for $1 < p < \infty$ (see Section 3.3 of Dung et al. (2018), Yanchenko (2020); Lizorkin & Nikol'skii (1990) and references therein). Therefore, the mixed frequency space can be viewed as an extension of the finite-dimensional mixed smooth Besov space to the infinite dimensional one. Approximation and estimation abilities of deep learning with ReLU activation for the true function in the finite-dimensional mixed smooth Besov space was investigated by Suzuki (2019) when $a_1 = \cdots = a_d$. They showed that deep learning can achieve the (near) minimax optimal rate in this setting and the mixed smoothness alleviates the curse of dimensionality.

## C.2 ANISOTROPIC SMOOTHNESS

Ingster & Stepanova (2006) considered a general function class represented by

$$\mathcal{F}_c := \left\{ f \in L^2([0,1]^\infty) : \sum_{l \in \mathbb{Z}_0^\infty} c_l^2 \langle f, \phi_l \rangle^2 < \infty \right\}$$

for a given sequence $c = (c_l)_{l \in \mathbb{Z}_0^\infty}$ with $c_l \in \mathbb{R}$. This class includes our $\gamma$-smooth space with $p = \theta = 2$. In particular, Ingster & Stepanova (2011) analyzed an estimation problem when $(c_l)_{l \in \mathbb{Z}_0^\infty}$ is given by

$$c_l^2 = \sum_{i=1}^{\infty} (2\pi l_i)^{2a_i},$$

for monotonically increasing sequence $a = (a_i)_{i=1}^{\infty} \in \mathbb{R}_{>0}^{\infty}$. It can be shown that, in this setting, $\mathcal{F}_c$ is specifically given by the following anisotropic Sobolev classe:

$$\mathcal{F}_c = \mathcal{W}_2^a := \left\{ f \in L^2([0,1]^\infty) : \sum_{i=1}^{\infty} \left\| \frac{\partial^{a_i} f}{\partial x_i^{a_i}} \right\|_2^2 < \infty \right\}.$$

This characterization highlights the intuition of the anisotropic smoothness, i.e., we assume the functions $f \in \mathcal{F}_c$ have different smoothness ($a_i$) with respect to each coordinate $x_i$. Here, by noticing that $\|\delta_s(f)\|_2^2 = \sum_{l \in \mathbb{Z}_0^\infty : \lfloor 2^{s_i-1} \rfloor \le |l_i| < 2^{s_i}} \langle f, \phi_l \rangle^2$, we have that

$$
\begin{aligned}
\|f\|_{\mathcal{W}_2^a}^2 = \sum_{l \in \mathbb{Z}_0^\infty} c_l^2 \langle f, \psi_l \rangle^2 &= \sum_{s \in \mathbb{N}_0^\infty} \sum_{l \in \mathbb{Z}_0^\infty : 2^{s_i-1} \le |l_i| < 2^{s_i}} c_l^2 \langle f, \psi_l \rangle^2 \\
&\ge \sum_{s \in \mathbb{N}_0^\infty} \max_{i \in \mathbb{N}} \{ (2\pi 2^{s_i-1})^{2a_i} \} \sum_{l \in \mathbb{Z}_0^\infty : 2^{s_i-1} \le |l_i| < 2^{s_i}} \langle f, \psi_l \rangle^2 \\
&\ge \sum_{s \in \mathbb{N}_0^\infty} 2^{2 \max\{a_i s_i\}_{i=1}^\infty} \sum_{l \in \mathbb{Z}_0^\infty : 2^{s_i-1} \le |l_i| < 2^{s_i}} \langle f, \psi_l \rangle^2 \\
&= \sum_{s \in \mathbb{N}_0^\infty} 2^{2 \max\{a_i s_i\}_{i=1}^\infty} \|\delta_s(f)\|_2^2 = \|f\|_{\mathcal{F}_{2,2}^\gamma}^2,
\end{aligned}
$$

for $\gamma(s) = \max_i\{a_i s_i\}$. Therefore, we see that the unit ball of the anisotropic Sobolev space $\mathcal{W}_2^a$ is included in the unit ball of $\mathcal{F}_{2,2}^\gamma$. Hence, our bounds in the following also give bounds for the anisotropic Sobolev space.

Suzuki & Nitanda (2021) also analyzed approximation and estimation error bounds of deep learning with the ReLU activation for the *anisotropic Besov spaces* defined on a finite dimensional space $[0,1]^d$ (Nikol'skii, 1975; Vybiral, 2006; Triebel, 2011).

## D    PROOF OF THEOREM 7

We can see that $\delta_s$ can be decomposed as

$$\delta_s(f)(x) = \sum_{k \in \mathbb{Z}_0^\infty : \lfloor 2^{s_i-1} \rfloor \leq |k_i| < 2^{s_i}} c_k \exp(2\pi i \langle k, x \rangle),$$

for $c_k \in \mathbb{C}$ and the imaginary number i. Thus, when $1 \leq p \leq 2$, from Theorem 1 of Nessel & Wilmes (1978), we have that

$$\|\delta_s(f)\|_2 \leq 2^{vs} \|\delta_s(f)\|_p, \tag{4}$$

where $v = (\frac{1}{p} - \frac{1}{2})_+$ and $2^{vs} = 2^{v \sum_{i=1}^\infty s_i}$. Furthermore, for $2 < p$, from the Cauchy-Schwarz inequality, we obtain

$$\begin{aligned}
\|\delta_s(f)\|_2^2 &= \int_{[0,1]^\infty} \delta_s(f)^2 d\lambda^\infty \\
&\leq \left(\int_{[0,1]^\infty} \delta_s(f)^p d\lambda^\infty\right)^{2/p} \left(\int_{[0,1]^\infty} 1 d\lambda^\infty\right)^{1-2/p} \\
&= \|\delta_s(f)\|_p^2.
\end{aligned}$$

Therefore, Eq. (4) holds all over the range of $1 \leq p < \infty$.

To show the approximation error in the assertion of the theorem, we consider approximating $f$ by $R_T(f)$ defined as

$$R_T(f) := \begin{cases} \sum_{s \in \mathbb{N}_0^\infty : \gamma(s) < T} \delta_s(f) & (1 \leq \theta \leq 2), \\ \sum_{s \in \mathbb{N}_0^\infty : \gamma'(s) < T} \delta_s(f) & (2 < \theta). \end{cases}$$

Then, we further approximate $R_T(f)$ by a fully connected neural network. For that purpose, we first analyze the approximation error $\|f - R_T(f)\|_2$ by $R_T(f)$. This can be evaluated in the following lemma.

**Lemma 17.** *Under the same setting as Theorem 7, we have that*

$$\|f - R_T(f)\|_2 \leq \begin{cases} 2^{-(1-v\alpha)T} \|f\|_{\mathcal{F}_{p,\theta}^\gamma} & (1 \leq \theta \leq 2), \\ 2^{-(1-v\alpha')T} \left[\sum_{T \leq \gamma'(s)} 2^{\frac{2\theta}{\theta-2}(\gamma'(s)-\gamma(s))}\right]^{1-2/\theta} \|f\|_{\mathcal{F}_{p,\theta}^\gamma} & (\theta > 2). \end{cases}$$

*Proof.* We show the inequality for the settings $1 \leq \theta \leq 2$ and $2 < \theta$ separately.

1. Approximation error $\|f - R_T(f)\|_2$ by $R_T(f)$ for $1 \leq \theta \leq 2$:
Using the orthogonality of $\delta_s$ between different $s$, we have

$$\begin{aligned}
\|f - R_T(f)\|_2^\theta &= (\|f - R_T(f)\|_2^2)^{\theta/2} \\
&\leq \left(\sum_{s \in \mathbb{N}_0^\infty : T \leq \gamma(s)} \|\delta_s(f)\|_2^2\right)^{\theta/2} \tag{5} \\
&\leq \sum_{s \in \mathbb{N}_0^\infty : T \leq \gamma(s)} (2^{vs} \|\delta_s(f)\|_p)^\theta \quad (\because \theta/2 \leq 1) \\
&= \sum_{s \in \mathbb{N}_0^\infty : T \leq \gamma(s)} (2^{\gamma(s)} 2^{vs-\gamma(s)} \|\delta_s(f)\|_p)^\theta, \tag{6}
\end{aligned}$$

where Eq. (4) is used to show Eq. (5). Then, for $s \in \mathbb{N}_0^\infty$ with $T \leq \gamma(s)$, using the assumption $v \frac{\sum_{i=1}^\infty s_i}{\gamma(s)} \leq v\alpha(\gamma) < 1$, we have that

$$2^{vs-\gamma(s)} \leq 2^{(v\alpha-1)\gamma(s)} \leq 2^{(v\alpha-1)T}.$$

Then, applying this inequality to Eq. (6), we obtain

$$\sum_{T \leq \gamma(s)} (2^{\gamma(s)} 2^{vs-\gamma(s)} \|\delta_s(f)\|_p)^\theta \leq 2^{-\theta(1-v\alpha)T} \sum_{T \leq \gamma(s)} \left(2^{\gamma(s)} \|\delta_s(f)\|_p\right)^\theta$$

$$\leq 2^{-\theta(1-v\alpha)T} \|f\|_{\mathcal{F}_{p,\theta}^\gamma}^\theta.$$

2. *Approximation error $\|f - R_T(f)\|_2$ by $R_T(f)$ for $2 < \theta$:*
Using again the orthogonality of $\delta_s(f)$ between different $s$, we obtain

$$\|f - R_T(f)\|_2^2 = \sum_{T \leq \gamma'(s)} \|\delta_s(f)\|_2^2,$$

and

$$\sum_{T \leq \gamma'(s)} \|\delta_s(f)\|_2^2 \leq \sum_{T \leq \gamma'(s)} (2^{vs} \|\delta_s(f)\|_p)^2$$

$$= \sum_{T \leq \gamma'(s)} \left(2^{vs-\gamma'(s)} 2^{\gamma'(s)-\gamma(s)} 2^{\gamma(s)} \|\delta_s(f)\|_p\right)^2. \tag{7}$$

Then, using the assumption $v\alpha' < 1$ for $\alpha' = \alpha(\gamma')$, we obtain $2^{vs-\gamma'(s)} \leq 2^{(v\alpha'-1)\gamma'(s)} \leq 2^{(v\alpha'-1)T}$. Applying this inequality to Eq. (7), we obtain

$$\sum_{T \leq \gamma'(s)} \left(2^{vs-\gamma'(s)} 2^{\gamma'(s)-\gamma(s)} 2^{\gamma(s)} \|\delta_s(f)\|_p\right)^2$$

$$\leq 2^{-2(1-v\alpha')T} \sum_{T \leq \gamma'(s)} \left(2^{\gamma'(s)-\gamma(s)} 2^{\gamma(s)} \|\delta_s(f)\|_p\right)^2.$$

Then, by using Cauchy-Schwarz inequality, we obtain

$$2^{-2(1-v\alpha')T} \sum_{T \leq \gamma'(s)} \left(2^{\gamma'(s)-\gamma(s)} 2^{\gamma(s)} \|\delta_s(f)\|_p\right)^2$$

$$\leq 2^{-2(1-v\alpha')T} \left[\sum_{T \leq \gamma'(s)} \left(2^{\gamma(s)} \|\delta_s(f)\|_p\right)^\theta\right]^{2/\theta} \times \left[\sum_{T \leq \gamma'(s)} \left(2^{\gamma'(s)-\gamma(s)}\right)^{2/(1-2/\theta)}\right]^{(1-2/\theta)}$$

$$\leq 2^{-2(1-v\alpha')T} \left[\sum_{T \leq \gamma'(s)} 2^{\frac{2\theta}{\theta-2}(\gamma'(s)-\gamma(s))}\right]^{1-2/\theta} \|f\|_{\mathcal{F}_{p,\theta}^\gamma}^2.$$

$\square$

Next, consider approximating $R_T$ by a neural network. We show it only for $1 \leq \theta \leq 2$, but the same argument can be applied for $2 < \theta$. Theorem 4.1 in Perekrestenko et al. (2018) asserts that there exist constants $C_1$, $C_2 > 0$ such that, for

$$L_{\tilde{\psi}} = C_1 \left[\left(\log \frac{1}{\epsilon}\right)^2 + \log\left(f_{\max}\right)\right],$$

there exists a neural network $\tilde{\psi}_{l_i} \in \Phi(L_{\tilde{\psi}}, 21, C_2, 21^2 L_{\tilde{\psi}})$ that can approximate $\psi_{l_i}$ as

$$\|\psi_{l_i} - \tilde{\psi}_{l_i}\|_{L^\infty([0,1])} \leq \epsilon.$$

Moreover, since $\psi_{l_i} \in [-\sqrt{2}, \sqrt{2}]$,

$$\| \max\{-\sqrt{2}, \min\{\sqrt{2}, \tilde{\psi}_{l_i}\}\} - \psi_{l_i} \|_{L^\infty([0,1])} \leq \epsilon. \tag{8}$$

Since, we can write

$$\max\{-\sqrt{2}, \min\{\sqrt{2}, x\}\} = \left[\eta(x) - \eta(x - \sqrt{2})\right] + \left[-\eta(-x) + \eta(-x + \sqrt{2})\right]$$

for $x \in \mathbb{R}$, it holds that $\max\{-\sqrt{2} \min\{\sqrt{2}, x\}\} \in \Phi(2, 4, \sqrt{2}, 16)$ and then we can see that

$$\max\{-\sqrt{2}, \min\{\sqrt{2}, \tilde{\psi}_{l_i}\}\} \in \Phi(L_{\tilde{\psi}} + 2, \ 21, \ \max\{C_2, \sqrt{2}\}, \ 21^2(L_{\tilde{\psi}} + 2)).$$

By using these facts, we can construct the neural network that takes value in $[-\sqrt{2}, \sqrt{2}]$. In the following, let

$$\hat{\psi}_{l_i} = \max\{-\sqrt{2}, \min\{\sqrt{2}, \tilde{\psi}_{l_i}\}\}, \ L_{\tilde{\psi}} = C_1 \left[\left(\log \frac{1}{\epsilon}\right)^2 + \log(f_{\max})\right] + 2.$$

Since we need to approximate a trigonometric polynomial, we need to (approximately) realize multiplications by neural networks. By Proposition 3 of Yarotsky (2017), using

$$L_\times = \left\lceil \log\left(\frac{3^{d_{\max}}}{\epsilon} + 5\right)\right\rceil \lceil \log d_{\max} \rceil, \ W_\times = 6 d_{\max}, \ S_\times = L_\times W_\times^2$$

and a constant $B_\times > 0$, there exists a neural network $\phi_\times \in \Phi(L_\times, W_\times, B_\times, S_\times)$ that satisfies

$$\left\| \phi_\times - \prod_{i=1}^{d_{\max}} x_i \right\|_{L^\infty([-1,1]^{d_{\max}})} \leq \epsilon.$$

Since $\hat{\psi}_{l_i} \leq \sqrt{2}$ is satisfied, we know that

$$\left\| \phi_\times\left(\frac{\hat{\psi}_{l_1}}{\sqrt{2}}, \ldots, \frac{\hat{\psi}_{l_{d_{\max}}}}{\sqrt{2}}\right) - \prod_{i=1}^{d_{\max}} \frac{\hat{\psi}_{l_i}}{\sqrt{2}} \right\|_{L^\infty([0,1]^\infty)} \leq \epsilon$$

$$\Rightarrow \left\| (\sqrt{2})^{d_{\max}} \phi_\times\left(\frac{\hat{\psi}_{l_1}}{\sqrt{2}}, \ldots, \frac{\hat{\psi}_{l_{d_{\max}}}}{\sqrt{2}}\right) - \prod_{i=1}^{d_{\max}} \hat{\psi}_{l_i} \right\|_{L^\infty([0,1]^\infty)} \leq \sqrt{2}^{d_{\max}} \epsilon. \tag{9}$$

Here, note that Eq. (8) yields that

$$\left\| \prod_{i=1}^{d_{\max}} \hat{\psi}_{l_i} - \prod_{i=1}^{d_{\max}} \psi_{l_i} \right\|_{L^\infty([0,1]^\infty)}$$

$$= \left\| \sum_{j=0}^{d_{\max}-1} \left( \prod_{i=1}^{j} \hat{\psi}_{l_i} \prod_{i=j+1}^{d_{\max}} \psi_{l_i} - \prod_{i=1}^{j+1} \hat{\psi}_{l_i} \prod_{i=j+2}^{d_{\max}} \psi_{l_i} \right) \right\|_{L^\infty([0,1]^\infty)}$$

$$= \left\| \sum_{j=0}^{d_{\max}-1} \left[ \prod_{i=1}^{j} \hat{\psi}_{l_i} \prod_{i=j+2}^{d_{\max}} \psi_{l_i} \left( \hat{\psi}_{l_{j+1}} - \psi_{l_{j+1}} \right) \right] \right\|_{L^\infty([0,1]^\infty)}$$

$$\leq \sqrt{2}^{d_{\max}} \sum_{j=0}^{d_{\max}-1} \left\| \hat{\psi}_{l_{j+1}} - \psi_{l_{j+1}} \right\|_{L^\infty([0,1]^\infty)}$$

$$\leq \sqrt{2}^{d_{\max}} d_{\max} \epsilon.$$

Then, by using triangle inequality and Eq. (9), we know that

$$\left\| (\sqrt{2})^{d_{\max}} \phi_\times\left(\frac{\hat{\psi}_{l_1}}{\sqrt{2}}, \ldots, \frac{\hat{\psi}_{l_{d_{\max}}}}{\sqrt{2}}\right) - \prod_{i=1}^{d_{\max}} \psi_{l_i} \right\|_{L^\infty([0,1]^\infty)}$$

$$\leq \left\| (\sqrt{2})^{d_{\max}} \phi_\times\left(\frac{\hat{\psi}_{l_1}}{\sqrt{2}}, \ldots, \frac{\hat{\psi}_{l_{d_{\max}}}}{\sqrt{2}}\right) - \prod_{i=1}^{d_{\max}} \hat{\psi}_{l_i} \right\|_{L^\infty([0,1]^\infty)} + \left\| \prod_{i=1}^{d_{\max}} \hat{\psi}_{l_i} - \prod_{i=1}^{d_{\max}} \psi_{l_i} \right\|_{L^\infty([0,1]^\infty)}$$

$$\leq (\sqrt{2})^{d_{\max}} (d_{\max} + 1)\epsilon.$$

Therefore, if we set

$$\hat{R}_T(f) := \sum_{\gamma(s)<T} \sum_{l\in J(s)} (\sqrt{2})^{d_{\max}} \langle f, \psi_l\rangle \phi_\times \left(\frac{\hat{\psi}_{l_1}}{\sqrt{2}}, \dots, \frac{\hat{\psi}_{l_{d_{\max}}}}{\sqrt{2}}\right),$$

where $J(s) := \{l \in \mathbb{Z}_0^\infty : \lfloor 2^{s_i-1}\rfloor \le |l_i| < 2^{s_i}\}$, we obtain that

$$\|\hat{R}_T - R_T\|_{L^\infty([-1,1]^d_{\max})}$$

$$= \left\|\sum_{s\in\mathbb{N}_0^\infty:\gamma(s)<T} \sum_{l\in J(s)} (\sqrt{2})^{d_{\max}} \langle f, \psi_l\rangle \left(\phi_\times\left(\frac{\hat{\psi}_{l_1}}{\sqrt{2}}, \dots, \frac{\hat{\psi}_{l_{d_{\max}}}}{\sqrt{2}}\right) - \frac{\psi_l}{\sqrt{2}^{d_{\max}}}\right)\right\|_{L^\infty([-1,1]^{d_{\max}})}$$

$$\le B_f G(T,\gamma)(\sqrt{2})^{d_{\max}}(d_{\max}+1)\epsilon,$$

where we used the fact $\langle f, \psi_l\rangle \le \|f\|_2 \le B_f$ in the last inequality. Hence, if we put

$$\epsilon = \frac{2^{-T}}{(\sqrt{2})^{d_{\max}} B_f (d_{\max}+1) G(T,\gamma)}, \tag{10}$$

then we have that

$$\|f - \hat{R}_T\|_2 \le \|f - R_T\|_2 + \|\hat{R}_T - R_T\|_{L^\infty([-1,1]^{d_{\max}})}$$

$$\le (2^{-T} + Q(T))\|f\|_{\mathcal{F}_{p,\theta}^\gamma},$$

where $Q(T) = \begin{cases} 2^{-(1-v\alpha)T} & (1 \le \theta \le 2), \\ 2^{-(1-v\alpha')T}\left[\sum_{T\le\gamma'(s)} 2^{\frac{2\theta}{\theta-2}(\gamma'(s)-\gamma(s))}\right]^{1-2/\theta} & (\theta > 2). \end{cases}$ Noting that

$2^{-T} \lesssim Q(T)$, we have that $\|f - \hat{R}_T\|_2 \lesssim Q(T)\|f\|_{\mathcal{F}_{p,\theta}^\gamma}$.

Finally, we evaluate the network size to achieve this approximation error. Since $\hat{R}_T$ is the linear combination of neural network $\phi_\times\left(\frac{\hat{\psi}_{l_1}}{\sqrt{2}}, \dots, \frac{\hat{\psi}_{l_{d_{\max}}}}{\sqrt{2}}\right)$, by putting

$$\begin{aligned} L &= L_{\hat{\phi}} + L_\times + 1, \\ W &= 21 d_{\max} G(T,\gamma), \\ S &= (21^2 d_{\max} L_{\hat{\phi}} + L_\times W_\times^2 + 1) G(T,\gamma), \\ B &= \max\{(\sqrt{2})^{d_{\max}} B_f, B_\times, C_2\}, \end{aligned}$$

we have that $\hat{R}_T \in \Phi(L, W, S, B)$. Substituting Eq. (10) to $L_\times$ and $L_{\hat{\phi}}$, we can evaluate as

$$L_{\hat{\psi}} = C_1 \left\lceil\left(\log\frac{1}{\epsilon}\right)^2 + \log(f_{\max}) + 2\right\rceil$$

$$= C_1 \left\lceil\left(T\log 2 + d_{\max}\log\sqrt{2} + \log S(\gamma,T) + \log B_f + \log d_{\max} + 1\right)^2 + \log f_{\max}\right\rceil$$

$$\le \left\lceil C_1 (6\max\{\log B_f, \log 2\})^2\right\rceil \left\lceil\left(\max\{d_{\max}^2, T^2, (\log S(\gamma,T))^2, \log f_{\max}, 2\}\right)\right\rceil$$

and

$$\begin{aligned} L_\times &= \left\lceil\log\left(\frac{3^{d_{\max}}}{\epsilon} + 5\right)\right\rceil \lceil\log d_{\max}\rceil \\ &\le \left\lceil\max\left\{\log\left(\frac{3^{d_{\max}}}{\epsilon}\right), \log 5\right\} + \log 2\right\rceil \lceil\log d_{\max}\rceil \\ &\le \lceil 6\max\{\log B_f, \log 5\}\rceil \lceil\max\{d_{\max}, T, \log G(T,\gamma)\}\rceil \lceil\log d_{\max}\rceil. \end{aligned}$$

Therefore, if we set $K = 2\max\left\{\lceil 6\max\{\log B_f, \log 5\}\rceil, \left\lceil C_1(6\max\{\log B_f, \log 2\})^2\right\rceil\right\}$, then it holds that

$$L \le 2K\max\left\{d_{\max}^2, T^2, (\log G(T,\gamma))^2, \log f_{\max}\right\},$$

and thus we also have

$$
\begin{aligned}
S &= (21^2 d_{\max} L_{\hat{\phi}} + L_{\times} W_{\times}^2 + 1) G(T, \gamma) \\
&\leq 2K(21^2 d_{\max} + 36 d_{\max}^2) \max\left\{d_{\max}^2, T^2, (\log G(T, \gamma))^2, \log f_{\max}\right\} G(T, \gamma) \\
&\leq 4 \times 21^2 K d_{\max}^2 \max\left\{d_{\max}^2, T^2, (\log G(T, \gamma))^2, \log f_{\max}\right\} G(T, \gamma).
\end{aligned}
$$

For the setting of $2 < \theta$, we can prove the bound by the same procedure. Thus, we obtain the assertion.

## E    PROOF OF THEOREM 9

Before we prove Theorem 9, we show the following lemma. The following lemma is inspired by Lemma 1.2 of Temlyakov (1983).

**Lemma 18.** *Suppose that $a = (a_i)_{i=1}^{\infty}$ and $a' = (a_i')_{i=1}^{\infty}$ are positive monotonically non-decreasing sequences such that $1 \leq a_1 = a_1'$ and*

$$
\prod_{i=2}^{\infty} \frac{1}{1 - 2^{-(a_i - a_1)}} < \infty, \quad \prod_{i=2}^{\infty} \frac{1}{1 - 2^{-\beta(a_i - a_i')}} < \infty,
$$

*for a positive constant $\beta > 0$. Then we have the following inequalities:*

$$
\sum_{s \in \mathbb{N}_0^{\infty} : \langle a', s \rangle \geq T} 2^{-\beta \langle a, s \rangle} \leq (1 - 2^{-\beta})^{-1} \left( \prod_{i=2}^{\infty} \frac{1}{1 - 2^{-\beta(a_i - a_i')}} \right) 2^{-\beta T} \tag{21}
$$

*and*

$$
\sum_{s \in \mathbb{N}_0^{\infty} : \langle a, s \rangle < T} 2^s \leq 8 \left( \prod_{i=2}^{\infty} \frac{1}{1 - 2^{-(a_i - a_1)}} \right) 2^T. \tag{22}
$$

*Proof.* First, note that

$$
\sum_{s \in \mathbb{N}_0^{\infty} : \langle a', s \rangle \geq T} 2^{-\beta \langle a, s \rangle}
$$

$$
= \left( \sum_{s_1=0}^{\infty} 2^{-\beta s_1} \right) \times \left( \sum_{(s_i)_{i=2}^{\infty} \in \mathbb{N}_0^{\infty} : \sum_{i=2}^{\infty} a_i' s_i \geq T} 2^{-\beta \sum_{i=2}^{\infty} a_i s_i} \right)
$$

$$
+ \sum_{(s_i)_{i=2}^{\infty} \in \mathbb{N}_0^{\infty} : \sum_{i=2}^{\infty} a_i' s_i < T} 2^{-\beta \sum_{i=2}^{\infty} a_i s_i} \left( \sum_{s_1 \in \mathbb{N} \cup \{0\} : s_1 \geq T - \sum_{i=2}^{\infty} a_i' s_i} 2^{-\beta s_1} \right). \tag{23}
$$

If $T$ satisfies $\sum_{i=2}^{\infty} a_i' s_i < T$, then we have that

$$
\sum_{s_1 \in \mathbb{N} \cup \{0\} : s_1 \geq T - \sum_{i=2}^{\infty} a_i' s_i} 2^{-\beta s_1} = \frac{2^{-\beta T} 2^{\beta \sum_{i=2}^{\infty} a_i' s_i}}{1 - 2^{-\beta}}.
$$

Thus, the second term of the right hand side of Eq. (23) can be evaluated as

$$
2^{-\beta T} (1 - 2^{-\beta})^{-1} \sum_{(s_i)_{i=2}^{\infty} \in \mathbb{N}_0^{\infty} : \sum_{i=2}^{\infty} a_i' s_i < T} 2^{-\beta \sum_{i=2}^{\infty} (a_i - a_i') s_i}.
$$

Next, we evaluate the first term of the right hand side of Eq. (23). We see that

$$
\sum_{s_1=0}^{\infty} 2^{-\beta s_1} = (1 - 2^{-\beta})^{-1},
$$

and

$$\sum_{(s_i)_{i=2}^{\infty}\in\mathbb{N}_0^{\infty}:\sum_{i=2}^{\infty}a'_i s_i\geq T} 2^{-\beta\sum_{i=2}^{\infty}a_i s_i}$$

$$\leq \sum_{(s_i)_{i=2}^{\infty}\in\mathbb{N}_0^{\infty}:\sum_{i=2}^{\infty}a'_i s_i\geq T} 2^{-\beta\sum_{i=2}^{\infty}(a_i-a'_i)s_i}2^{-\beta\sum_{i=2}^{\infty}a'_i s_i}$$

$$\leq 2^{-\beta T}\sum_{(s_i)_{i=2}^{\infty}\in\mathbb{N}_0^{\infty}:\sum_{i=2}^{\infty}a'_i s_i\geq T} 2^{-\beta\sum_{i=2}^{\infty}(a_i-a'_i)s_i}.$$

Thus the first term of the right hand side of Eq. (23) can be bounded by

$$2^{-\beta T}(1-2^{-\beta})^{-1}\sum_{(s_i)_{i=2}^{\infty}\in\mathbb{N}_0^{\infty}:\sum_{i=2}^{\infty}a'_i s_i\geq T} 2^{-\beta\sum_{i=2}^{\infty}(a_i-a'_i)s_i}.$$

Hence, by noticing that

$$\sum_{s\in\mathbb{N}_0^{\infty}} 2^{-\beta\sum_{i=2}^{\infty}(a_i-a'_i)s_i} = \prod_{i=2}^{\infty}\left(\sum_{s_i=0}^{\infty}2^{-\beta(a_i-a'_i)s_i}\right) = \prod_{i=2}^{\infty}\frac{1}{1-2^{-\beta(a_i-a'_i)}},$$

and combining the evaluations above, Eq. (23) yields that

$$\sum_{s\in\mathbb{N}_0^{\infty}:\langle a',s\rangle\geq T} 2^{-\beta\langle a,s\rangle} \leq 2^{-\beta T}(1-2^{-\beta})^{-1}\prod_{i=2}^{\infty}\frac{1}{1-2^{-\beta(a_i-a'_i)}},$$

which yields the first inequality.

Next, we show the second inequality. We decompose $\sum_{s\in\mathbb{N}_0^{\infty}:\langle a,s\rangle<T} 2^s$ as $\sum_{t=0}^{T}I_t$ where $I_t$ is defined as

$$I_t := \sum_{s\in\mathbb{N}_0^{\infty}:t-1\leq\langle a,s\rangle<t} 2^{\sum_{i=1}^{\infty}s_i}.$$

Here, for $1\leq t\leq T$, we can evaluate $I_t$ as

$$I_t \leq 2^{2t}\sum_{s\in\mathbb{N}_0^{\infty}:t-1\leq\langle a,s\rangle<t} 2^{\sum_{i=1}^{\infty}s_i-2\langle a,s\rangle}$$

$$\leq 2^{2t}\sum_{s\in\mathbb{N}_0^{\infty}:t-1\leq\langle a,s\rangle<t} 2^{-\sum_{i=1}^{\infty}(2a_i-1)s_i}.$$

Since $a_i\geq 1$, we have $a_i\leq 2a_i-1$ $(i\geq 1)$ and thus by Eq. (21) with $a'_i=a_1$ $(\forall i\in\mathbb{N})$ and $\beta=1$, we have

$$\sum_{s\in\mathbb{N}_0^{\infty}:t-1\leq\langle a,s\rangle<t} 2^{-\sum_{i=1}^{\infty}(2a_i-1)s_i} \leq \sum_{s\in\mathbb{N}_0^{\infty}:t-1\leq\langle a,s\rangle<t} 2^{-\langle a,s\rangle}$$

$$\leq \sum_{s\in\mathbb{N}_0^{\infty}:t-1\leq\langle a',s\rangle} 2^{-\langle a,s\rangle}$$

$$\leq 4\left(\prod_{i=2}^{\infty}\frac{1}{1-2^{-(a_i-a_1)}}\right)2^{-t},$$

where we used the first assertion (21) in the last inequality. Therefore, we can obtain

$$I_t \leq 2^{2t}\sum_{s\in\mathbb{N}_0^{\infty}:t-1\leq\langle a,s\rangle<t} 2^{-\sum_{i=1}^{\infty}(2a_i-1)s_i} \leq 4\left(\prod_{i=2}^{\infty}\frac{1}{1-2^{-(a_i-a_1)}}\right)2^{t}.$$

Now using this bound, we can verify that

$$\sum_{s\in\mathbb{N}_0^{\infty}:\langle a,s\rangle<T} 2^s \leq 8\left(\prod_{i=2}^{\infty}\frac{1}{1-2^{-(a_i-a_1)}}\right)\sum_{t=0}^{T}2^t$$

$$\leq 8\left(\prod_{i=2}^{\infty}\frac{1}{1-2^{-(a_i-a_1)}}\right)2^T,$$

which yields the second assertion (22). $\qquad\square$

Now, we are ready to show Theorem 9. First, we give the proof in the setting of $\gamma(s) = \langle a, s \rangle$.

**Proof for mixed smoothness ($\gamma(s) = \langle a, s \rangle$).** First, we consider the setting $1 \le \theta \le 2$. Lemma 18 yields

$$G(T, \gamma) = \sum_{s \in \mathbb{N}_0^\infty : \langle a, s \rangle < T} 2^s = \sum_{s \in \mathbb{N}_0^\infty : \langle \frac{a}{a_1}, s \rangle < \frac{T}{a_1}} 2^s \le 8 \left( \prod_{i=2}^\infty \frac{1}{1 - 2^{\frac{-(a_i - a_1)}{a_1}}} \right) 2^{\frac{T}{a_1}}. \qquad (24)$$

Moreover, we can easily see that

$$\alpha = \sup_{s \in \mathbb{N}_0^\infty} \frac{\sum_{i=1}^\infty s_i}{\langle a, s \rangle} = \frac{1}{a_1},$$

because $a$ is a positive monotonically non-decreasing sequence. By the assumption $a_i = \Omega(i^q)$,

$$d_{\max} \sim T^{1/q}, \quad f_{\max} \sim T$$

are satisfied. Now, by using the filter $w \in \mathbb{R}^{C \times 1 \times W'}$ with the width $W' = d_{\max}$, the number of output channels $C = d_{\max}$ and the number of input channels $C' = 1$ given by

$$w_{i,1,j} = \begin{cases} 1 & (i = j), \\ 0 & (i \ne j), \end{cases}$$

for $i, j \in [d_{\max}]$, we can see that

$$(\mathrm{Conv}_{1,w}(X))_1 = \begin{pmatrix} x_1 \\ \vdots \\ x_{d_{\max}} \end{pmatrix}.$$

By Theorem 7, if we set

$$L = 2K \max \left\{ T^{\frac{2}{q}}, T^2 \right\},$$

$$W = 21 \left( \prod_{i=2}^\infty \frac{1}{1 - 2^{\frac{-(a_i - a_1)}{a_1}}} \right) T^{\frac{1}{q}} 2^{\frac{T}{a_1}},$$

$$S = 1764K \left( \prod_{i=2}^\infty \frac{1}{1 - 2^{\frac{-(a_i - a_1)}{a_1}}} \right) T^{\frac{2}{q}} \max \left\{ T^{\frac{2}{q}}, T^2 \right\} 2^{\frac{T}{a_1}},$$

$$B = (\sqrt{2})^{d_{\max}} K',$$

where $K, K' > 0$ are constants, for any function $f \in U(\mathcal{F}_{p,\theta}^{\langle a, s \rangle})$, there exists a neural network $\hat{R}_T \in \Phi(L, W, S, B)$ such that

$$f'(X) := \hat{R}_T(x_1, \ldots, x_{d_{\max}})$$

satisfies

$$\|f' - f\|_2 \le 2^{-(1 - \frac{v}{a_1})T}.$$

Since $f'(X) = \left( \hat{R}_T \circ \mathrm{Conv}_{1,w}(X) \right)_1$, we can see that $f'$ is a dilated CNN: $f' \in \mathcal{P}(L', B', W', C, L, W, B, S)$ where $L' = 1$, $B' = 1$ and $W' = C = d_{\max}$.

Next, we consider the setting $2 < \theta$. Let $a'_1 = \frac{a_1}{2}$, and for $\delta = a_2 - a_1$ (which is positive by the assumption $a_2 > a_1$) and a constant $u$ that satisfies $2 < u < 2 + \frac{2\delta}{a_1}$, we set $a'_i = \frac{a_i}{u}$ for $i \ge 1$. Then, $a'_1 < a'_2 \le \ldots$ is satisfied, that is, $a' = (a'_i)_{i=1}^\infty$ is a positive monotonically increasing sequence. Moreover, by the assumption $a_i = \Omega(i^q)$, it holds that, for all $c > 0$,

$$\prod_{i=2}^\infty \frac{1}{1 - 2^{-c\left( \frac{a_i}{a_1} - \frac{2a'_i}{a_1} \right)}} = \prod_{i=2}^\infty \frac{1}{1 - 2^{-c\left( \frac{1}{a_1} - \frac{2}{ua_1} \right)a_i}} < \infty. \qquad (25)$$

By noticing that $\frac{2(a_i' - a_i)}{a_1} = 2\left(\frac{1}{u} - 1\right)\frac{a_i}{a_1} \le -\frac{a_i}{a_1}$ where we used $u > 2$, we also have that

$$\sum_{s \in \mathbb{N}_0^\infty : \langle a', s \rangle \ge T} 2^{\frac{2\theta}{\theta-2}\langle a'-a, s \rangle} \le \sum_{s \in \mathbb{N}_0^\infty : \langle 2a'/a_1, s \rangle \ge 2T/a_1} 2^{-\frac{a_1\theta}{\theta-2}\langle a/a_1, s \rangle}.$$

Moreover, since we have verified Eq. (25), we can apply Lemma 18 to the right hand side of this inequality and it can be further bounded as

$$\sum_{s \in \mathbb{N}_0^\infty : \langle 2a'/a_1, s \rangle \ge 2T/a_1} 2^{-\frac{a_1\theta}{\theta-2}\langle a/a_1, s \rangle} \le 2^{-\frac{2\theta}{\theta-2}T}.$$

By setting $\gamma'(s) = \langle a', s \rangle$ and using Theorem 7, we can see that if we define the set of CNN $\mathcal{P}$ by the same argument as the case of $1 \le \theta \le 2$, for all $f \in U(\mathcal{F}_{p,\theta}^\gamma)$, there exists a dilated CNN $f' \in \mathcal{P}$ such that

$$\|f' - f\|_2 \lesssim 2^{-(1-v\alpha(\gamma'))T}(2^{-\frac{2\theta}{\theta-2}T})^{1/2-1/\theta} = 2^{-2(1-\frac{v}{a_1})T}$$

is satisfied, where we used $\alpha(\gamma') = \frac{2}{a_1}$ by the definition of $a'$. Here, as in Eq. (24), we have that

$$G(T, \gamma') \le 8\left(\prod_{i=2}^\infty \frac{1}{1 - 2^{\frac{-(a_i' - a_1')}{a_1'}}}\right) 2^{\frac{T}{a_1'}} \le 8\left(\prod_{i=2}^\infty \frac{1}{1 - 2^{\frac{-(a_i - a_1)}{a_1}}}\right) 2^{\frac{2T}{a_1}}.$$

Therefore, by resetting $T \leftarrow 2T$, we obtain the same result as in $1 \le \theta \le 2$.

**Proof for anisotropic smoothness** ($\gamma(s) = \max_i\{a_i s_i\}_i$). Here, we consider the setting of mixed smoothness $\gamma(s) = \max_i\{a_i s_i\}_i$. We note that

$$\alpha(\gamma) = \sup_{s \in \mathbb{N}_0^\infty} \frac{\sum_{i=1}^\infty s_i}{\sup_i\{a_i s_i\}} \le \sup_{s \in \mathbb{R}_{>0}^\infty} \frac{\sum_{i=1}^\infty s_i}{\sup_i\{a_i s_i\}}$$

$$= \sup_{T>0} \sup_{\{s \in \mathbb{R}_{>0} : \max_i\{a_i s_i\} = T\}} \frac{\sum_{i=1}^\infty s_i}{T}.$$

The condition $\sup_i\{a_i s_i\} \le T$ is equivalent to the condition that $s_i \le \frac{T}{a_i}$ for all $i \in \mathbb{N}$, the right hand side can be further bounded as

$$\alpha(\gamma) \le \sum_{i=1}^\infty \frac{1}{a_i}.$$

Therefore, in the setting of $1 \le \theta \le 2$, Theorem 7 with $\alpha = \sum_{i=1}^\infty \frac{1}{a_i}$ yields that

$$\|R_T(f) - f\|_2 \le 2^{-(1-\delta\alpha)T}\|f\|_{\mathcal{F}_{p,\theta}^\gamma}.$$

Since it holds that

$$\sum_{s \in \mathbb{N}_0^\infty : \gamma(s) < T} 2^s \le \prod_{i=1}^\infty \left(\sum_{s_i=0}^{\lceil \frac{T}{a_i} \rceil} 2^{s_i}\right) \le 2^{\sum_{i=1}^\infty \lceil \frac{T}{a_i} \rceil},$$

$G(T, \gamma) \lesssim 2^{\sum_{i=1}^\infty \lceil \frac{T}{a_i} \rceil}$ is satisfied. By noticing that $a$ is a positive monotonically increasing sequence with polynomial order growth, the same argument as in the setting of $\gamma(s) = \langle a, s \rangle$ can be applied. Then, we obtain the assertion.

## F  PROOF OF THEOREM 10

The estimation error can be derived by evaluating the bias and variance trade-off. The bias can be evaluated by the approximation error which has been analyzed in the previous sections, and the variance can be characterized by the complexity of the model. As a measure of complexity of the model, we utilize the covering number of the model (van der Vaart & Wellner, 1996).

**Definition 19** (Covering number). *For a norm space $\mathcal{F}$ equipped with a norm $\|\cdot\|$, the $\epsilon$-covering number $\mathcal{N}(\mathcal{F}, \epsilon, \|\cdot\|)$ the minimum number of balls with radius $\epsilon$ (measured by the norm $\|\cdot\|$) to cover the norm space $\mathcal{F}$:*

$$\mathcal{N}(\mathcal{F}, \delta, \|\cdot\|) := \inf\{n \in \mathbb{N} : \exists(f_1, \ldots, f_n) \in \mathcal{F}^n, \ \forall f \in \mathcal{F}, \ \exists i \in [n], \ \|f_i - f\| \leq \epsilon\}.$$

The covering number of the model of the dilated CNNs can be evaluated as in the following lemma.

**Lemma 20.** *The log-covering number of the set of the dilated CNNs $\mathcal{P}(L', B', W', C', L, W, B, S)$ can be bounded as*

$$\log \mathcal{N}(\mathcal{P}, \delta, \|\cdot\|_\infty) \lesssim (S + W'C')(L + L') \log\left(\frac{LL'(B' \vee 1)(B \vee 1)C'W'W}{\delta}\right).$$

*Proof.* The assertion can be shown by evaluating how strongly a small perturbation of parameters can deviate the function. For $w \in \mathbb{R}^{C' \times W'}$ and $h \in \mathbb{N}$, we have that

$$\|w \star_h X - w' \star_h X\|_\infty \leq \|(w - w') \star_h X\|_\infty \leq W'C\|w - w'\|_\infty\|X\|_\infty,$$
$$\|w \star_h X - w \star_h X'\|_\infty \leq W'C'\|w\|_\infty\|X - X'\|_\infty.$$

Therefore, for $F, F' \in \mathbb{R}^{C' \times C' \times W'}$, we have

$$\|\mathrm{Conv}_{h,F} \circ X - \mathrm{Conv}_{h,F'} \circ X\|_\infty = \max_{i \in [C']} \|(F_{i,:,:} \star_h X) - (F'_{i,:,:} \star_h X)\|_\infty$$

$$\leq W'C' \left(\max_{i \in [C']} \|F_{i,:,:} - F'_{i,:,:}\|_\infty\right) \|X\|_\infty$$

$$= W'C'\|F - F'\|_\infty\|X\|_\infty, \qquad (26)$$

$$\|\mathrm{Conv}_{h,F} \circ X - \mathrm{Conv}_{h,F} \circ X'\|_\infty \leq \max_{i \in [C']} \|(F_{i,:,:} \star_h X) - (F_{i,:,:} \star_h X')\|$$

$$\leq W'C' \left(\max_{i \in [C']} \|F_{i,:,:}\|_\infty\right) \|X - X'\|_\infty$$

$$\leq W'C'\|F\|_\infty\|X - X'\|_\infty. \qquad (27)$$

Here, for CNNs $f, g$ such that

$$f(X) = \mathrm{Conv}_{W'^{L'}, F_{L'}} \circ \cdots \circ \mathrm{Conv}_{W'^l, F_l} \circ \cdots \circ \mathrm{Conv}_{1, F_1} \circ X,$$
$$g(X) = \mathrm{Conv}_{W'^{L'}, F'_{L'}} \circ \cdots \circ \mathrm{Conv}_{W'^l, F'_l} \circ \cdots \circ \mathrm{Conv}_{1, F'_1} \circ X,$$

we define

$$\mathcal{A}_l(f)(X) := \mathrm{Conv}_{W'^{l-1}, F_{l-1}} \circ \cdots \circ \mathrm{Conv}_{1, F_1} \circ X,$$
$$\mathcal{B}_l(g)(X') := \mathrm{Conv}_{W'^{L'}, F'_{L'}} \circ \cdots \circ \mathrm{Conv}_{W'^{l+1}, F'_{l+1}} \circ X',$$

where $\mathcal{A}_1(f)(X) = \mathcal{B}_{L'}(f)(X) = X$. By using these notation and the triangle inequality, we can see that it holds that

$$\|f(X) - g(X)\|_\infty$$

$$\leq \sum_{l=1}^{L'} \|\mathcal{B}_l(g) \circ \mathrm{Conv}_{W'^l, F_l} \circ \mathcal{A}_l(f)(X) - \mathcal{B}_l(g) \circ \mathrm{Conv}_{W'^l, F'_l} \circ \mathcal{A}_l(f)(X)\|_\infty.$$

Hence, by applying the inequalities (27) and (26), we can see that

$$\|f(X) - g(X)\|_\infty \leq L'(W'C')^{L'}\|X\|_\infty \left(\prod_{i=1}^{L'} \|F_i\|_\infty\right) \max_{l=1,\ldots L'} \|F_l - F'_l\|_\infty$$

$$\leq L'(W'C'B')^{L'} \max_{l=1,\ldots L'} \|F_l - F'_l\|_\infty.$$

Now, for $f_{\mathrm{FNN}} \in \Phi(L, W, B, S)$, we have

$$|f_{\mathrm{FNN}}(x) - f_{\mathrm{FNN}}(x')| \leq (BW)^L\|x - x'\|_\infty,$$

and thus we also have

$$\|f_{\text{FNN}} \circ f(X) - f_{\text{FNN}} \circ g(X)\|_\infty \le L'(BW)^L (TC'B')^{L'} \max_{l=1,\dots L'} \|F_l - F'_l\|_\infty.$$

On the other hands, $f_{\text{FNN}}, g_{\text{FNN}} \in \Phi(L, W, B, S)$ can be represented as

$$f_{\text{FNN}}(x) = (A_L \eta(\cdot) + b_L) \circ \cdots \circ (A_l \eta(\cdot) + b_l) \circ \cdots \circ \dots (A_1 x + b_1),$$
$$g_{\text{FNN}'}(x) = (A'_L \eta(\cdot) + b_L) \circ \cdots \circ (A'_l \eta(\cdot) + b'_l) \circ \cdots \circ \dots (A'_1 x + b'_1).$$

Then, by noticing that $\|g(X)\|_\infty \le (W'C')^{L'}$, the same argument as Lemma 3 of Suzuki (2019) yields that, if

$$\max_{l=1,\dots,L} \max\{\|A_l - A'_l\|_\infty, \|b_l - b'_l\|_\infty\} \le \delta, \tag{28}$$

then the $L_\infty$ distance between the composite functions $f_{\text{FNN}} \circ g$ and $g_{\text{FNN}'} \circ g$ can be bounded as

$$\|f_{\text{FNN}} \circ g(X) - g_{\text{FNN}'} \circ g(X)\|_\infty \le \delta L(W'C')^{L'} \{(B+1)(W+1)\}^L.$$

Since the triangle inequality yields

$$\|\text{FNN} \circ f(X) - \text{FNN}' \circ g(X)\|_\infty \le \|\text{FNN} \circ f(X) - \text{FNN} \circ g(X)\|_\infty + \|\text{FNN} \circ g(X) - \text{FNN}' \circ g(X)\|_\infty,$$

by combining these inequalities, under the condition (28) and $\max_{l=1,\dots L'} \|F_l - F'_l\|_\infty \le \delta$, we obtain

$$\|\text{FNN} \circ f(X) - \text{FNN}' \circ g(X)\|_\infty$$
$$\le \delta L'(BW)^L (W'C'B')^{L'} + \delta L(W'C')^{L'} \{(B+1)(W+1)\}^L.$$

Therefore, by noticing that the number of combinations of non-zero parameter configurations in $\Phi(L, W, B, S)$ is bounded by $(W+1)^{LS}$, we can again utilize the same argument as Lemma 3 of Suzuki (2019) to we obtain

$$\log \mathcal{N}(\mathcal{P}, \delta, \|\cdot\|_\infty) \lesssim (S + W'C')(L + L') \log\left(\frac{LL'(B' \vee 1)(B \vee 1)C'W'W}{\delta}\right).$$

$\square$

Using a covering number bound of a model, a standard analysis of the ERM estimator gives the following bound on its estimation error.

**Lemma 21** (Theorem 2.6 of Hayakawa & Suzuki (2020); Schmidt-Hieber (2020))**.** *Let $\widehat{f}$ be any ERM estimator that takes its value in a model $\mathcal{F} \subset L^\infty([0,1]^\infty)$. Suppose that there exists a constant $F > 0$ such that $\|f^\circ\|_\infty \le F$ and $\|f\| \le F$ for any $f \in \mathcal{F}$. Then for any $0 < \delta < 1$ satisfying $\mathcal{N}(\mathcal{F}, \delta, \|\cdot\|_\infty) \ge 3$, it holds that*

$$\mathrm{E}_{P^n}[\|\widehat{f} - f^\circ\|^2_{P_X}] \le 4 \inf_{f \in \mathcal{F}} \|f - f^\circ\|^2_{P_X} + C\left((F + \sigma)\frac{\mathcal{N}(\mathcal{F}, \delta, \|\cdot\|_\infty)}{n} + \delta(F + \sigma)\right),$$

*where $C$ is a universal constant.*

Since we have assumed $P_X$ is absolutely continuous to the Lebesgue measure $\lambda^\infty$ and $\|\frac{dP_X}{d\lambda^\infty}\|_\infty < \infty$, $\|\cdot\|_{P_X}$ in the right hand side can be replaced by $\|\cdot\|_2$ with a constant factor multiplication. Now, we are ready to show Theorem 10.

**Proof of Theorem 10** First, we consider the setting of $\gamma(s) = \langle a, s \rangle$. By Theorem 9, by setting

$$L' = 1,$$
$$B' = 1,$$
$$W' \sim T^{\frac{1}{q}},$$
$$C' \sim T^{\frac{1}{q}},$$
$$L \sim \max\left\{T^{\frac{2}{q}}, T^2\right\},$$
$$W \sim \left(\prod_{i=2}^{\infty} \frac{1}{1 - 2^{\frac{-(a_i - a_1)}{a_1}}}\right) T^{\frac{1}{q}} 2^{\frac{T}{a_1}},$$
$$S \sim \left(\prod_{i=2}^{\infty} \frac{1}{1 - 2^{\frac{-(a_i - a_1)}{a_1}}}\right) T^{\frac{2}{q}} \max\left\{T^{\frac{2}{q}}, T^2\right\} 2^{\frac{T}{a_1}},$$
$$B \sim (\sqrt{2})^{T^{\frac{1}{q}}},$$

we have that, for any $f^\circ \in U(\mathcal{F}_{p,\theta}^\gamma)$, there exists a dilated CNN $f \in \mathcal{P}(L', B', W', C', L, W, B, S)$ such that

$$\|f - f^\circ\|_2^2 \leq 2^{-2(1 - v/a_1)T}.$$

Moreover, by Lemma 20, the covering number of $\mathcal{P}$ can be bounded as

$$\log\left(\mathcal{N}(\mathcal{P}, \delta, \|\cdot\|_\infty)\right) \lesssim \left(\prod_{i=2}^{\infty} \frac{1}{1 - 2^{\frac{-(a_i - a_1)}{a_1}}}\right) 2^{\frac{T}{a_1}} T^{\frac{2}{q}+1} \max\left\{T^{\frac{4}{q}}, T^4\right\} \log\left(\frac{T}{\delta}\right).$$

Here, under the assumption $\|f^\circ\|_\infty \leq B_f$, we can also obtain that there exists $f \in \bar{\mathcal{P}}(B_f, L', B', W', C', L, W, B, S)$ such that $\|f - f^\circ\|_2^2 \leq 2^{-2(1 - v/a_1)T}$, and a covering number evaluation $\log\left(\mathcal{N}(\bar{\mathcal{P}}, \delta, \|\cdot\|_\infty)\right) \leq \log\left(\mathcal{N}(\mathcal{P}, \delta, \|\cdot\|_\infty)\right)$. Therefore, by Lemma 21, the ERM estimator $\widehat{f}$ taking its value in $\bar{\mathcal{P}}(B_f, L', B', W', C, L, W, B, S)$ satisfies

$$\mathrm{E}_{P^n}[\|\widehat{f} - f^\circ\|_{P_X}^2]$$

$$\lesssim 2^{-2(1 - \frac{v}{a_1})T} + (B_f^2 + \sigma^2)\frac{\left(\prod_{i=2}^{\infty} \frac{1}{1 - 2^{\frac{-(a_i - a_1)}{a_1}}}\right) 2^{\frac{T}{a_1}} T^{\frac{2}{q}+1} \max\left\{T^{\frac{4}{q}}, T^4\right\} \log(\frac{T}{\delta})}{n} + \delta(B_f + \sigma).$$

Thus, by taking $T$ to satisfy $2^{\frac{T}{a_1}} = n^{\frac{1}{2(a_1 - v)+1}}$ and letting $\delta = \frac{1}{n}$, we obtain that

$$\mathrm{E}_{P^n}[\|\widehat{f} - f^\circ\|_{P_X}^2] \lesssim \left(\prod_{i=2}^{\infty} \frac{1}{1 - 2^{\frac{-(a_i - a_1)}{a_1}}}\right) n^{-\frac{2(a_1 - v)}{2(a_1 - v)+1}} (\log n)^{\frac{2}{q}+2} \max\{(\log n)^{\frac{4}{q}}, (\log n)^4\},$$

which yields the first assertion.

Next, we consider the setting of $\gamma = \max_i\{a_i s_i\}_i$. We again apply Theorem 9 so that, by setting

$$L' = 1,$$
$$B' = 1,$$
$$W' \sim T^{\frac{1}{q}},$$
$$C' \sim T^{\frac{1}{q}},$$
$$L \sim \max\left\{T^{\frac{2}{q}}, T^2\right\},$$
$$W \sim T^{\frac{1}{q}} 2^{T/\tilde{a}},$$
$$S \sim T^{\frac{2}{q}} \max\left\{T^{\frac{2}{q}}, T^2\right\} 2^{T/\tilde{a}},$$
$$B \sim (\sqrt{2})^{T^{\frac{1}{q}}},$$

we have that, for any $f^\circ \in U(\mathcal{F}_{p,\theta}^\gamma)$, there exists a dilated CNN $f \in \mathcal{P}(L', B', W', C', L, W, B, S)$ such that

$$\|f - f^\circ\|_2^2 \lesssim 2^{-2(1-v/\tilde{a})T}.$$

Therefore, by the same argument as the setting of mix smoothness, we can verify

$$\mathrm{E}_{P^n}[\|\widehat{f} - f^\circ\|_{P_X}^2] \lesssim^{-\frac{2(\tilde{a}-v)}{2(\tilde{a}-v)+1}} (\log n)^{\frac{2}{q}+2} \max\{(\log n)^{\frac{4}{q}}, (\log n)^4\},$$

which yields the second assertion.

## G    Proofs of Theorems 13 and 14

First, we prove the following lemma.

**Lemma 22.** *Let* $X = (x_i)_{i=1}^\infty \in \mathbb{R}^\infty$. *For any sequence* $(x_{i_1}, \ldots, x_{i_N})$ *satisfying* $i_j \leq H^{L'}$ *with arbitrary integer* $H > 1$ *and* $i_1 < i_2 < \cdots < i_N$, *there exists a dilated CNN where the number of layers is* $L'$, *the width of filters is* $H$, *the number of channels of each layer is* $N$ *and the filters* $W_l \in \mathbb{R}^{N \times N \times H}$ $(2 \leq l \leq T)$ *and* $W_1 \in \mathbb{R}^{N \times H}$ *in each layer satisfy* $\|W_l\|_\infty \leq 1$ $(l = 1, \ldots, L')$ *such that*

$$(x_{i_1}, \ldots, x_{i_N})^\top = \left(\mathrm{Conv}_{H^{L'-1}, W_{L'}} \circ \cdots \circ \mathrm{Conv}_{H^{l-1}, W_l} \circ \cdots \circ \mathrm{Conv}_{1, W_1} \circ X\right)_1.$$

*Proof.* Let

$$\mathcal{A}_l(X) = \mathrm{Conv}_{H^{l-1}, W_l} \circ \cdots \circ \mathrm{Conv}_{1, W_1} \circ X.$$

First, we let $(W_1)_{k,j} = 1$ if $(i_k - 1 \bmod T) = j - 1$ and $(W_1)_{k,j} = 0$ otherwise, for $k \in [N]$ and $j \in [T]$. Then, for each $k \in [N]$, there exists $i_k^{(1)} \in [H^{L'-1}]$ such that $\mathcal{A}_1(X)_{k,(i_k^{(1)}-1)H+1} = x_{i_k}$, indeed, $i_k^{(1)}$ can be obtained as $i_k^{(1)} = \lfloor (i_k - 1)/H \rfloor + 1$. In the following, we recursively define $W_l$ and $i_k^{(l)}$ for $l = 2, \ldots, L'$ so that

$$\mathcal{A}_l(X)_{k,(i_k^{(l)}-1)H^l+1} = x_{i_k} \quad (k \in [N]) \tag{29}$$

is satisfied. We remark that this is satisfied for $l = 2$. Suppose that, for $l$, we have $W_l$ and $i_k^{(l)}$ that satisfy the condition (29). If we define

$$\mathcal{A}'_{l,i:}(X) := \left(\mathcal{A}_l(X)_{:,(i-1)H^{l+1}+(j-1)H^l+1}\right)_{j=1}^\infty \in \mathbb{R}^{N \times \infty}$$

for $i \in \mathbb{N}$, then we have that

$$\mathcal{A}_{l+1}(X)_{:,(i-1)H^{l+1}+1} = (\mathrm{Conv}_{H^l, W_{l+1}} \circ \mathcal{A}_l(X))_{:,(i-1)H^{l+1}+1}$$

$$= \begin{pmatrix} (W_{l+1})_{1,:,:} \star_1 \mathcal{A}'_{l,i:}(X) \\ \vdots \\ (W_{l+1})_{N,:,:} \star_1 \mathcal{A}'_{l,i:}(X) \end{pmatrix}_1.$$

Therefore, by letting $i_k^{(l+1)}$ as $i_k^{(l+1)} = \lfloor (i_k^{(l)} - 1)/H \rfloor + 1$ (i.e., the integer such that $(i_k^{(l+1)} - 1)H + 1 \leq i_k^{(l)} < i_k^{(l+1)}H$) and setting $W_{l+1}$ so that $(W_{l+1})_{k,j} = 1$ if $(i_k^{(l)} - 1 \bmod T) = j - 1$ and $(W_{l+1})_{k,j} = 0$ otherwise $(k \in [N]$ and $j \in [T])$, we can see that the condition (29) $(\mathcal{A}_{l+1}(X)_{k,(i_k^{(l+1)}-1)T^{l+1}+1} = x_{i_k}$ for all $k \in [N])$ holds for $l + 1$. Therefore, by the inductive argument, we have that

$$\mathcal{A}_{L'}(X)_{:,1} = (x_{i_k})_{k=1}^N,$$

because $i_k^{(L')}$ satisfies $0 \leq i_k^{(L')} - 1 \leq (i_k - 1)/H^{L'}$ which yields $i_k^{(L')} = 1$ for all $k \in [N]$ by the assumption $i_k \leq H^{L'}$. $\qquad\square$

A pictorial illustration of the proof of Lemma 22 with $H = W'$, $N = C'$ and $L' = 2$ is given in Figure 1.

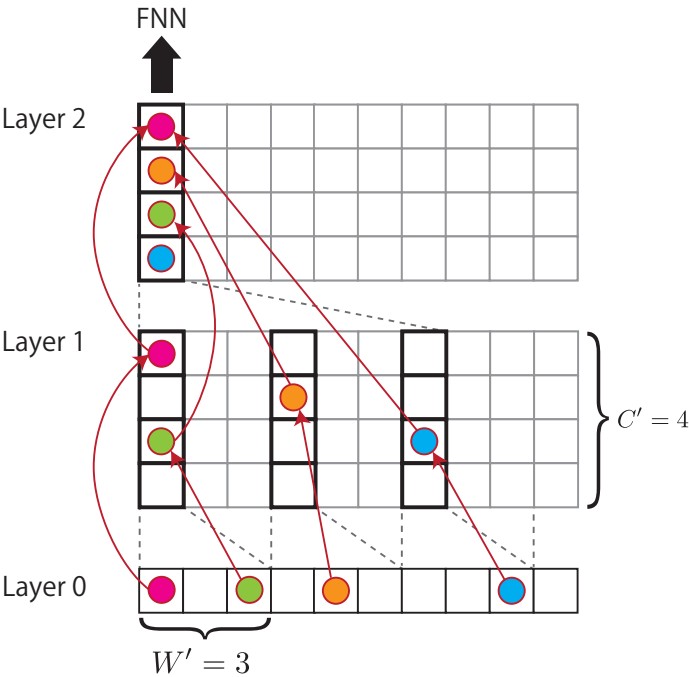

Figure 1: Illustration of how important features are extracted by the dilated convolution. The colored circles indicate the important features $(x_{i_1}, \ldots, x_{i_{C'}})$.

*Proofs of Theorems 13 and 14.*

Let $(i_j)_{j=1}^\infty$ be the sequence of indices that yields $a_{i_1} < a_{i_2} < \cdots$ for the given positive sequence $a = (a_i)_{i=1}^\infty$. We only give the proof for the mixed smoothness setting $\gamma(s) = \langle a, s \rangle$. As for the anisotropic smoothness setting, the same reasoning can be applied.

Since $\|a\|_{wl^q} \le 1$ is satisfied by the assumption, we have $a_{i_j} \ge j^q$ for all $j \in \mathbb{N}$. Therefore, we can verify that for all $T > 0$, it holds that

$$
\begin{aligned}
I(T, \gamma) &= \{i : \exists s \in \mathbb{N}_0^\infty,\ s_i \ne 0,\ \langle a, s \rangle \le T\} \\
&= \left\{ i_j : 1 \le j \le T^{\frac{1}{q}} \right\}.
\end{aligned}
$$

Hence, we can see that $d_{\max} = T^{\frac{1}{q}}$, $f_{\max} = T^{\frac{1}{q}}$ by their definitions. Since $a_i = \Omega(\log i)$, there exists a constant $Q > 0$ such that

$$
a_i \ge Q \log i
$$

for all $i \in [N]$. Then, if $a_i \le T$, $i \le \exp\left(\frac{T}{Q}\right)$ is satisfied. By using Lemma 22 with $N = d_{\max}$, we can construct the dilated convolutional structure

$$
\mathrm{Conv}_{3^{\lceil \frac{T}{Q} \rceil - 1}, W_{\lceil \frac{T}{Q} \rceil}} \circ \cdots \circ \mathrm{Conv}_{1, W_1} \circ X
$$

that extracts all of the elements in $I(T, \gamma)$. In particular, for a fully connected neural network $\hat{R}_T$ with $d_{\max}$ dimensional input, we have that

$$
\left( \hat{R}_T \circ \mathrm{Conv}_{3^{\lceil \frac{T}{Q} \rceil - 1}, W_{\lceil \frac{T}{Q} \rceil}} \circ \cdots \circ \mathrm{Conv}_{1, W_1} \circ X \right)_1 = \hat{R}_T((x_i : i \in I(T, \gamma))), \tag{32}
$$

that is, the convolutional neural network in the right hand side depends only on the elements in $I(T, \gamma)$.

On the other hand, by Lemma 18, we have that

$$\sum_{s\in\mathbb{N}_0^\infty:\langle a,s\rangle<T} 2^s = \sum_{s\in\mathbb{N}_0^\infty:\sum_{j=1}^\infty a_{i_j}s_{i_j}<T} 2^s$$

$$\lesssim \left(\prod_{i\neq i_1} \frac{1}{1-2^{-\frac{(a_i-a_{i_1})}{a_{i_1}}}}\right) 2^{\frac{T}{a_{i_1}}}.$$

Therefore, by Theorem 7, if we set

$$L \sim \max\{T^{\frac{2}{q}}, T^2\},\ W \sim 21\left(\prod_{i\neq i_1}\frac{1}{1-2^{-\frac{(a_i-a_{i_1})}{a_{i_1}}}}\right)T^{\frac{1}{q}}2^{\frac{T}{a_{i_1}}},$$

$$S \sim \left(\prod_{i\neq i_1}\frac{1}{1-2^{-\frac{(a_i-a_{i_1})}{a_{i_1}}}}\right)\max T^{\frac{2}{q}}\{T^{\frac{2}{q}},T^2\}2^{\frac{T}{a_{i_1}}},\ B \sim \sqrt{2}^{T^{\frac{1}{q}}},$$

there exists a neural network $\hat{R}_T \in \Phi(L,W,B,S)$ such that $f'(X) = \hat{R}_T((x_i : i \in I(T,\gamma))$ satisfies

$$\|f'-f^o\|_2 \leq 2^{-(1-\frac{v}{a_{i_1}})T}. \tag{35}$$

Therefore, by combining Eq. (32) and Eq. (35), we can set

$$L' = \left\lceil\frac{T}{Q}\right\rceil \sim T,\ W' = 3,\ C' = T^{\frac{1}{q}},\ B' = 1,$$

and $(L,W,S,B)$ as above so that there exists $f' \in \mathcal{P}(L',B',W',C',L,W,B,S)$ such that

$$\|f'-f\|_2 \leq 2^{-(1-\frac{v}{a_{i_1}})T}.$$

This yields the proof of Theorem 13.

The remaining proof for the estimation error (Theorem 14) can be done in the same manner as that of Theorem 10, which yields Theorem 14. □

## H NUMERICAL EXPERIMENTS

### H.1 THE EXPERIMENT OF DIMENSION INDEPENDENCE

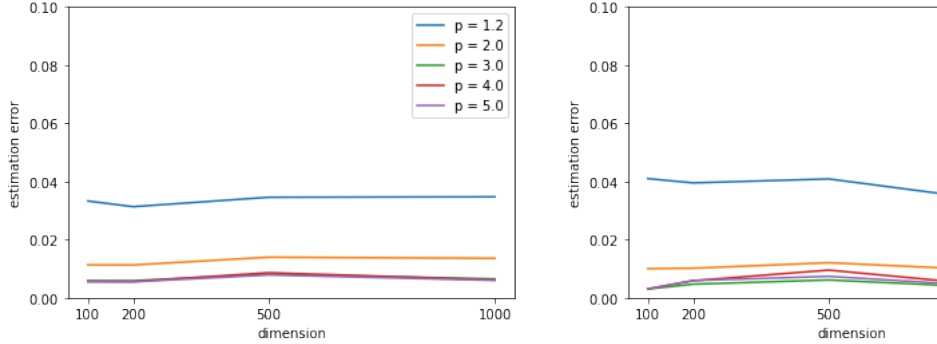

Figure 2: Smoothness with polynomial order increase

Figure 3: Smoothness with sparsity

In this section, we verify the dimension independence of CNNs by numerical experiments using functions with anisotropic smoothness. The experiments were conducted in the following settings:

1. Smoothness with Polynomial order increase:
   We consider the following function as the true function:

$$f^\circ(x) = \sum_{k=1}^{d} \frac{\sqrt{2}^k}{\sum_{i=1}^{k} 2^{i^p}} \prod_{i=1}^{k} \cos\left(2\pi x_i\right),$$

where $p, d \in \mathbb{N}$ are parameters. Data generation model: $y_i = f^\circ(x_i)$ $(i = 1, \ldots, n)$. Parameter settings: $n = 128$, $p = 1.2, 2.0, 3.0, 4.0, 5.0$, $d = 100, 200, 500, 1000$. The trained model: CNN with the kernel width $W' = 10$, the number of channels $C' = 10$, and the depth $L' = 1$.

2. Smoothness with sparse structure:
   We consider the following function as the true function:

$$f^\circ(x) = \sum_{k=1}^{[d/10]} \frac{\sqrt{2}^k}{\sum_{i=1}^{k} 2^{i^p}} \prod_{i=1}^{k} \cos\left(2\pi x_{10i}\right),$$

where $p, d \in \mathbb{N}$ are parameters. Data generation model: $y_i = f^\circ(x_i)$ $(i = 1, \ldots, n)$. Parameter setting: $n = 128$, $p = 1.2, 2.0, 3.0, 4.0, 5.0$, $d = 100, 200, 500, 1000$. The trained model: Dilated CNN with the depth $L' = 3$ layers, the kernel width $W' = 3$, and the number of channels $C' = 10$.

In each experiment, we estimated $f^\circ$ from $n$ observations $(x_i, y_i)_{i=1}^{n}$ using a CNN or dilated CNN. The results are shown in Figure 2 and Figure 3. These results show that for the same $p$, we don't see any deterioration of the estimation error by increasing the dimensionality $d$ of the input. These results are consistent with the fact that the theoretical upper bound is dimension independent and depends only on smoothness. Moreover, in the setting of sparse smoothness, we again observe that the estimation error is dimension-independent by using dilated CNN, which is also consistent with our theoretical findings.

## I    EXTENSION OF 2D CONVOLUTION

Here, we present that a similar argument can be applied to 2D convolution for image inputs. For that purpose, we consider a quite simple setting where the true function $f^\circ$ has a mixed/anisotropic smoothness with respect to the wavelet coefficients of the input image. More precisely, suppose that the input image $X$ has size $N \times N$ where $N = 2^K$ for an integer $K$ ($X \in [0, 1]^{N \times N}$) and let $(\alpha_{k,j})_{j=1}^{3(N/2^k)^2}$ for $k \in K$ be the wavelet coefficient at the level $k$. Then, we assume the true function is $\gamma$-smooth with respect to the wavelet coefficients $(\alpha_{1,1}, \alpha_{1,2}, \ldots, \alpha_{1,3(N/2)^2}, \alpha_{2,1}, \ldots, \alpha_{2,3(N/2^2)}, \ldots, \alpha_{K,1}) \in \mathbb{R}^{N^2}$. Usually, $N^2$ is large and a standard analysis suffers from the curse of dimensionality.

The construction of wavelet coefficients is as follows (see Daubechies (1992) for more details). The wavelet transform is obtained based on a pair of a *wavelet filter* $\varphi \in \mathbb{R}^N$ and *scaling filter* $\psi \in \mathbb{R}^N$. We assume this pair of filters are given. For $u \in \mathbb{R}^{M_1}$ and $v \in \mathbb{R}^{M_2}$, let

$$u \otimes v := \begin{pmatrix} u_1 v_1 & \cdots & u_1 v_{M_2} \\ \vdots & \ddots & \vdots \\ u_{M_1} v_1 & \cdots & u_{M_1} v_{M_2} \end{pmatrix} \in \mathbb{R}^{M_1 \times M_2},$$

and for $x, y \in \mathbb{R}^{M_1 \times M_2}$, let

$$\langle x, y \rangle := \sum_{i=1}^{M_1} \sum_{j=1}^{M_2} x_{i,j} y_{i,j}.$$

The level $k$ wavelet and scaling bases are obtained by

$$\varphi_i^{(k)} = \sum_{j=1}^{2^{k-1}} \varphi_{i+(j-1)N/2^{k-1}}, \quad \psi_i^{(k)} = \sum_{j=1}^{2^{k-1}} \varphi_{i+(j-1)N/2^{k-1}} \quad (i = 1, \ldots, N/2^{k-1}).$$

We apply a shift operation on these basis functions as
$$\varphi^{(k,j)} = \left( \varphi^{(k)}_{[(i-1+2(j-1)) \bmod (N/2^{k-1})]+1} \right)_{i=1}^{N/2^{k-1}} \in \mathbb{R}^{N/2^{k-1}} \quad (j = 1, \ldots, N/2^k),$$
and we define $\psi^{(k,j)}$ in the same way. Using these filter, we can calculate the wavelet coefficients in an inductive manner. At the first level, we obtain
$$\alpha^{(1,1)}_{i,j} = \langle \varphi^{(1,i)} \otimes \varphi^{(1,j)}, X \rangle \quad (i, j \in [N/2]),$$
$$\alpha^{(1,2)}_{i,j} = \langle \varphi^{(1,i)} \otimes \psi^{(1,j)}, X \rangle \quad (i, j \in [N/2]),$$
$$\alpha^{(1,3)}_{i,j} = \langle \psi^{(1,i)} \otimes \varphi^{(1,j)}, X \rangle \quad (i, j \in [N/2]),$$
$$\alpha^{(1,4)}_{i,j} = \langle \psi^{(1,i)} \otimes \psi^{(1,j)}, X \rangle \quad (i, j \in [N/2]).$$
Let $A^{(1,q)} = (\alpha^{(1,q)}_{i,j})_{i,j}$ for $q = 1, 2, 3, 4$. At the $k$-th level, we assume that we have already have $A^{(k-1,q)} \in \mathbb{R}^{N/2^{k-1}}$ ($q \in [4]$). Then, the $k$-th level coefficients can be obtained by
$$\alpha^{(k,1)}_{i,j} = \langle \varphi^{(k,i)} \otimes \varphi^{(k,j)}, A^{(k-1,1)} \rangle \quad (i, j \in [N/2^k]),$$
$$\alpha^{(k,2)}_{i,j} = \langle \varphi^{(k,i)} \otimes \psi^{(k,j)}, A^{(k-1,1)} \rangle \quad (i, j \in [N/2^k]),$$
$$\alpha^{(k,3)}_{i,j} = \langle \psi^{(k,i)} \otimes \varphi^{(k,j)}, A^{(k-1,1)} \rangle \quad (i, j \in [N/2^k]),$$
$$\alpha^{(k,4)}_{i,j} = \langle \psi^{(k,i)} \otimes \psi^{(k,j)}, A^{(k-1,1)} \rangle \quad (i, j \in [N/2^k]).$$
We again let $A^{(k,q)} = (\alpha^{(k,q)}_{i,j})_{i,j}$ for $q = 1, 2, 3, 4$. Then, we obtain the wavelet coefficient as $A(X) = (A^{(k,2)}, A^{(k,3)}, A^{(k,4)})_{k=1}^{K-1}$ in a recursive manner that represents strength of each frequency component $k$ of the input image $X$ at each location $j$. We assume that the true function $f^{\circ}$ is $\gamma$-smooth with respect to this coefficient (more precisely its vectorization $\mathrm{vec}(A(X)) \in \mathbb{R}^{N^2}$).

**Assumption 23.** *The true function $f^{\circ} : \mathbb{R}^{N \times N} \to N$ is $\gamma$-smooth with respect to $\mathrm{vec}(A(X))$. The smoothness satisfies the sparsity (Assumption 12). Moreover, we also assume Assumptions 3 is satisfied.*

From now on, we construct a 2D-CNN structure to extract variables in $I(T, \gamma)$ from $\mathrm{vec}(A(X))$. Suppose that the support of $\varphi$ and $\psi$ are included in the first $H$ components, then the inner product appeared above are executable just in a local $H \times H$ region. For example, the Haar wavelet has $H = 2$. Therefore, the inner product to extract the coefficients can be realized by the standard convolution operation in CNNs.

From now on, we employ the same notations used in the proof of Theorems 13 and 14. We $T > 0$ be any positive real. We define $(d_{\max}, f_{\max}, G)$ as in Theorem 7. If we can show that a CNN can extract variables included in $I(T, \gamma)$, then we can apply the same argument to Theorems 13 and 14.

Let the number of channels of of 2D convolution be $C_k = d_{\max} + 4$ for $k = 1, \ldots, K$ and $C_0 = 1$. For each layer $k \in [K-1]$, we will define a filter $W^{(k)} \in \mathbb{R}^{C_k \times C_{k-1} \times H_k \times H_k}$ where $H_k = \min\{H, N/2^k\}$. Let $\mathcal{A}_k(X) \in \mathbb{R}^{C_k \times N/2^k \times N/2^k}$ be the output from the $k$-th layer of the CNN. Then, $\mathcal{A}_k(X)$ is updated by the following convolution with the interval $h = 2$: $\mathcal{A}_0(X) = X$ and
$$\mathcal{A}_k(X)_{q,i,j} = \sum_{i_1=1}^{H_k} \sum_{j_1=1}^{H_k} \sum_{q_1=1}^{C_{k-1}} \mathcal{A}_{k-1}(X)_{q_1,2(i-1)+i_1,2(j-1)+j_1} W^{(k)}_{q,q_1,i_1,j_1} \quad (q \in C_k, \ i, j \in [N/2^k]),$$
where, by convention, we let $\mathcal{A}_{k-1}(X)_{i,j} = \mathcal{A}_{k-1}(X)_{(i-1 \bmod N/2^{k-1})+1,(j-1 \bmod N/2^{k-1})+1}$ for $i$ and $j$ which are out of the range $\{1, \ldots, N/2^{k-1}\}$.

Next, we show that an appropriately defined filter can extract $I(T, \gamma)$. First, we let a part of $W^{(k)}$ as follows to extract $(A^{(k,1)}, A^{(k,2)}, A^{(k,3)}, A^{(k,4)})$:
$$W^{(k)}_{1,1,:,:} = \varphi_{1:H_k} \otimes \varphi_{1:H_k},$$
$$W^{(k)}_{2,1,:,:} = \varphi_{1:H_k} \otimes \psi_{1:H_k},$$
$$W^{(k)}_{3,1,:,:} = \psi_{1:H_k} \otimes \varphi_{1:H_k},$$
$$W^{(k)}_{4,1,:,:} = \psi_{1:H_k} \otimes \psi_{1:H_k},$$

where $x_{i:i'} = (x_i, x_{i+1}, \ldots, x_{i'})^\top$ for a vector $x$.

Second, for $k \geq 2$, we extract the "important coefficients" included in $I(T, \gamma)$ from $(A^{(k-1,2)}, A^{(k-1,3)}, A^{(k-1,4)})$ that are already extracted in the last layer. For that purpose, let $W^{(k)}_{j,q,i_1,j_1} = 1$ for $2 \leq q \leq 4$, $j \geq 5$ and $i_1, j_1 \in [2]$ if $A^{(k-1,q)}$ contains the $(j-4)$-th element of $I(T, \gamma)$ at its $(i_1, j_1)$-th component. Otherwise, let $W^{(k)}_{j,q,i_1,j_1} = 0$.

Finally, we extract the important coefficients included in $\mathcal{A}_{k-1}(X)_{q,:,:}$ for $q \geq 5$. For $5 \leq q \leq N$, let $W_{j,q,i_1,i_2} = 1$ for $j \geq 5$ and $i_1, j_1 \in [2]$ if $\mathcal{A}_{k-1}(X)_{q,:,:}$ contains the $(j-4)$-th element of $I(T, \gamma)$ at its $(i_1, j_1)$-th component. Otherwise, we let $W_{j,q,i_1,i_2} = 0$.

Since the interval of convolution is $h = 2$, the size of $\mathcal{A}_K(X)$ is $C_K \times 1 \times 1$ which can be seen as a "vector." By the construction above, it contains all coefficients included in $I(T, \gamma)$. Therefore, by feeding $\tilde{X} = \text{vec}(\mathcal{A}_K(X))$ to an FNN $\hat{R}_T$ as considered in the proof of Theorems 13 and 14, we obtain the same approximation and estimation error bounds as those theorems under Assumption 23. Here, note that the number of parameters in the CNN constructed above has the same order as the 1D-situation thanks. Thus, we obtain the same convergence rate as Theorems 13 and 14.

