# OpenReview forum: "Learnability of convolutional neural networks for infinite dimensional input via mixed and anisotropic smoothness"
_ICLR.cc/2022/Conference — ICLR 2022 Spotlight_

### Official Review · Reviewer_BeQE · 2021-11-02

**Correctness:** 4
**Technical Novelty And Significance:** 4
**Empirical Novelty And Significance:** Not applicable
**Recommendation:** 8
**Confidence:** 5

**Main Review:**

The setting with an infinite dimensional input space is a very interesting topic and is closely related to distribution regression. The rates of approximation and estimation for least squares regression by deep ReLU neural networks given in the paper are nice contributions to the theoretical study of deep learning. I recommend the paper to be accepted.

The authors might discuss the following issues:

1. It would help the readers if the authors can point out the connection between the topic discussed in the paper and distribution regression. Though the setting with an infinite dimensional input space is valuable for describing some learning problems, it would improve the usage of the results if the implementation of deep neural networks in this setting could be discussed.

2. For CNNs, the authors propose only linear maps for the convolutional layers without activation and argue in Remark 5 that their analysis can be applied straightforwardly when ReLU is used. This is arguable and the authors might give more details. To my opinion, the rates of approximation and learning in the paper are derived mainly by using the last fully connected layer.

3. A theory for approximation and learning by 1-D CNNs with vector input spaces has been developed recently. The authors should mention some of the existing results in this development.



**Summary Of The Paper:**

The authors consider approximation and learning by deep neural networks in the setting with an infinite dimensional input space. They provide nice rates for approximating and learning functions with mixed or anisotropic smoothness. The networks studied in the paper include fully connected ReLU networks and those generated by linear convolutional layers followed by a fully connected layer.

**Summary Of The Review:**

The setting with an infinite dimensional input space is very interesting. The rates of approximation and estimation by deep ReLU neural networks given in the paper are nice.

---

> ### Author Response · Authors · 2021-11-22
> **Reply to reviewer BeQE**
>
> Thank you for your valuable feedback which is quite informative for us to improve the manuscript. Please find our answer in the following.
>
> **1. It would help the readers if the authors can point out the connection between the topic discussed in the paper and distribution regression.**
>
> Indeed, the distribution regression is one of important examples for an infinite dimensional input problem. All methods for the distribution regression extracts an infinite dimensional feature vector for each input distribution (e.g., basis function expansion of the density function), therefore by putting a feature extraction network before the CNN of our method, then the output from the feature extraction network (which is high dimensional) can be fed into the CNNs or FNNs constructed in our paper. Therefore, our results can be applied to the distribution regression problem with a small modification, i.e., adding feature extraction module from the input distribution.
>
>
> **2. To my opinion, the rates of approximation and learning in the paper are derived mainly by using the last fully connected layer.**
>
> That is right. In our analysis, the convolution layer is used only to extract important features and thus the linear operation required in the convolution layer is just multiplication of a sparse matrix with entries 0 or 1 which can be realized by ReLU activation network. In that sense, the the derived rate is mainly owing to the last fully connected layer. However, the feature extraction layer is quite important because it can cover exponentially large range of inputs. More practical functional of CNN is also analyzed in 2D convolution example in Appendix I where the convolution layer works as the wavelet decomposition of the input image.
>
>
> **3. A theory for approximation and learning by 1-D CNNs with vector input spaces has been developed recently. The authors should mention some of the existing results in this development.**
>
> We are not sure which paper you intended exactly. One of the recent approximation and learning error analysis for CNNs is given by [R1]. The relation to this work is mentioned in Section 3. [R1] analyzed learning ability of CNNs with a ResNet structure in a classification task where the data are distributed on a low-dimensional manifold and established a rate which only depends on the dimensionality of the  low  dimensional  manifold. However, the input should be distributed on a low dimensional manifold, while our analysis allows its support to be infinite dimensional.
>
> [R1] Liu, M. Chen, T. Zhao, and W. Liao.   Besov function approximation and binary classification on low-dimensional manifolds using convolutional residual networks. InProceedings of the 38thInternational Conference on Machine Learning, volume 139 ofProceedings of Machine LearningResearch, pp. 6770–6780. PMLR, 2021.

---

### Official Review · Reviewer_iiDe · 2021-11-03

**Correctness:** 4
**Technical Novelty And Significance:** 4
**Empirical Novelty And Significance:** Not applicable
**Recommendation:** 8
**Confidence:** 3

**Main Review:**

I find the paper very interesting and novel, in that it explores adaptivity of convolutional architectures to useful smoothness structure in the target, providing dimension-independent rates which may hold even for infinite-dimensional signals.
I am thus in favor of acceptance. Nevertheless, the paper is quite dense with notation, and I believe the clarity and presentation of the paper could be significantly improved:

* the definition of gamma-smoothness was a bit hard to digest, and would benefit from more intuition, perhaps relating to penalizing derivatives, if appropriate.

* the motivation about being more/less sensitive to high vs low frequencies in an image could be discussed further. In particular, the authors suggest that the $x_i$ for large $i$ could correspond to higher frequencies, but in practice these probably just correspond do different spatial locations of pixels. In this sense, the final result showing adaptivity to different orderings of $a_i$ for dilated CNNs is much more compelling than the previous ones.

* more comments on the requirements for the network architectures would be helpful, since they are quite heavy to read in the current presentation. For instance, it would be helpful to mention in the beginning of section 5.2 that the CNNs only use *one* convolutional layer here. In Definition 4, it would be helpful to make a remark that you're choosing h = W', and why.

* generally, more intuition on the results and their proofs, particularly why CNNs enable the adaptivity they do, would be helpful. More discussions after the main theorems, e.g. on limitations of FNNs and benefits of CNNs, would be welcome as well.

A question: would using strided convolutions (with stride = filter size) instead of dilated convolutions also allow for similar adaptivity?

Minor remarks:
- Table 1: what is meant by $d \ll n$? do such rates only hold asymptotically?
- Table 1: notation-wise, using $\tilde a$ to denote its inverse would perhaps make more sense?
- Section C.1 contains a "Comment: Needs more precise references"

**Summary Of The Paper:**

The paper studies non-parametric regression for functions defined on infinite-dimensional input data (such as signals in $\ell^2$), using fully-connected networks or dilated convolutional networks (in the CNN case, convolutional layers are followed by a fully-connected network).
The authors consider certain smoothness classes similar to mixed or anisotropic smoothness but extended to infinite-dimensional input data, which requires per-coordinate smoothness orders ($a_i$ in Definition 2) that are non-decreasing or increasing with some rate.
For such classes, the authors obtain rates that only depend on $a_1$ for the mixed smoothness case, or $\sum_i a_i^{-1}$ for the anisotropic case, for the ERM under certain classes of FNNs or CNNs. Notably, CNNs avoid the need to selecting specific finite subsets of variables as inputs (needed by FNNs), and dilated CNNs additionally allow some adaptivity to sparsity in the $a_i$, in particular, by avoiding dependence on the specific order of the $a_i$ (the growth condition is on the sorted values instead of the unsorted ones for the non-dilated case with a single convolutional layer).

**Summary Of The Review:**

A good paper tackling an interesting question, namely studying adaptivity of convolutional architectures to certain function spaces.

---

> ### Author Response · Authors · 2021-11-22
> **Reply to reviewer iiDe**
>
> Thank you for your valuable feedback which is helpful for us to improve the manuscript. Please find our answer in the following.
>
> **1. the definition of gamma-smoothness was a bit hard to digest, and would benefit from more intuition, perhaps relating to penalizing derivatives, if appropriate.**
>
> The definition would look heavy at the first glance, but we also think that the definition of gamma-smoothness is already intuitive because it explicitly imposes penalties over different frequency components. As for the connection to the penalizing derivatives, there is indeed close connection. Actually, we have given an explanation about connection to a penalty on smoothness toward each coordinate direction as a special setting of mixed smoothness in Appendix C.2. Please look at the section for more details.
>
> **2. the motivation about being more/less sensitive to high vs low frequencies in an image could be discussed further. ... the final result showing adaptivity to different orderings of ai for dilated CNNs is much more compelling than the previous ones.**
>
> We agree with your opinion. That is why we added included Theorems 13 and 14. Moreover, we also included the 2D convolution analysis as an application of our theories in Appendix I. This analysis also highlights an ability of CNNs to extract informative features distributed over different spatial locations.
>
>
> **3. more comments on the requirements for the network architectures would be helpful, ... For instance, it would be helpful to mention in the beginning of section 5.2 that the CNNs only use one convolutional layer here. In Definition 4, it would be helpful to make a remark that you're choosing h = W', and why.**
>
> We have added one sentence in the beginning of Section 5.2 to explain that only one layer CNN is used there. We also added a footnote in Definition 4 that h is chosen as a power of W' and it is useful to show the feature extraction ability of CNNs. We added an illustration of CNNs in Figure 1 (Appendix G), which also illustrates why h=W'^k in Definition 4 can achieve feature extraction.
>
>
> **generally, more intuition on the results and their proofs, particularly why CNNs enable the adaptivity they do, would be helpful.**
>
> We have added an illustration about how the CNN structure can adaptively extract informative features in Figure 1 (Appendix G), which we believe is helpful to capture the intuition. Moreover, we added the table of notations to relieve the issue of heavy notations.
> Unfortunately, there is no extra space in the main text. We may produce more spaces by reducing several mathematical contents but we would like to avoid it because it would produce inaccurate description of the results. Instead, we think that the main message can be properly understood by the readers even in the current form because we see that the reviewers have correctly understood the main result of the paper.
>
>
> **A question: would using strided convolutions (with stride = filter size) instead of dilated convolutions also allow for similar adaptivity?**
>
> Yes, that is true. The strided convolution corresponds to neglecting the redundant output from the dilated convolution that are skipped in the next layer's dilated convolution.
> Therefore, we may utilize the strided convolution instead of the dilated convolution to obtain the same result.
>
>
> **Minor comments:**
> - **Table 1: what is meant by d << n? do such rates only hold asymptotically?**
> The bound holds a non-asymptotic setting, but the bound is meaningful (i.e., sufficiently small) only when the sample size is sufficiently large compared with the dimension $d$ because the constant hidden in the rate depends on the dimensionality $d$.
> - **Table 1: notation-wise, using $\tilde{a}$ to denote its inverse would perhaps make more sense?**
> We agree with your opinion. We have fixed the definition as you suggested.
> - **Section C.1 contains a "Comment: Needs more precise references"**
> Thank you very much for your kindly pointing it out. We have overlooked that there remains a comment for editing. We have removed it.

---

### Official Review · Reviewer_DfK7 · 2021-11-05

**Correctness:** 4
**Technical Novelty And Significance:** 4
**Empirical Novelty And Significance:** Not applicable
**Recommendation:** 8
**Confidence:** 3

**Main Review:**

The paper investigated the approximation error and estimation error of convolutional neural networks in the function spaces with mixed smoothness and anisotropic smoothness. It was shown that the convergence rates are determined by the smoothness of the target functions and independent of input dimension under mild conditions, which theoretically supports the practical success of convolutional neural networks and provides insights on how to avoid the curse of dimensionality by deep learning methods. These results are novel and of interest. I have some minor concerns as follows.

1. Are the obtained rates also minimax optimal when $1\leq p<2$?

2. Is it possible to extend the theoretical results (Theorems 9 and 10) to the case when $L'$ depends on $T$ and $L_1$ (or $L_2$) is fixed?


**Summary Of The Paper:**

The paper proves novel dimension-independent bounds for both approximation and estimation by convolutional neural networks when the input is infinite-dimensional and the target function has mixed and anisotropic smoothness. Moreover, the authors show the advantages of dilated convolution when the smoothness of the target function has a sparse structure.

**Summary Of The Review:**

Overall, the paper is well-written and technically sound. The results should be of great interest to the theoretical deep learning community.

---

> ### Author Response · Authors · 2021-11-22
> **Reply to reviewer DfK7**
>
> Thank you for your suggestive comments. Please find our answer in the following.
>
> **1. Are the obtained rates also minimax optimal when $1 \leq p < 2$?**
>
> Thank you very much for noticing an important point. Indeed, we believe that it is *not* minimax optimal and it can be improved. More precisely, the term $v$ could be removed (in exchange for some additional poly-log(n) order). This is true for the finite dimensional setting, but due to infinite dimensionality, it is quite non-trivial to show the optimal rate in our setting. This is mainly because there appears infinitely many cross terms compared with the finite dimensional setting. However, we believe that the minimax optimal rate could be achieved (possibly, with some additional poly-log(n) order). We would like to defer it to the future work.
>
>
> **2. Is it possible to extend the theoretical results (Theorems 9 and 10) to the case when L' depends on T and L1 (or L2) is fixed?**
>
> Thanks for your interesting suggestion. We think it is possible. Indeed, [R1] showed equivalence between convolutional networks and fully connected networks. Although [R1] analyzed a finite dimensional setting, the idea could be imported to our setting. We think the practical CNNs are also doing like that, that is, the CNN layers work nonlinear function approximation in addition to feature extractions.
>
> [R1] Petersen, P. and Voigtlaender, F. Equivalence of approximation by convolutional neural networks and fully connected networks. Proceedings of the American Mathematical Society 148 (4), 1567--1581, 2020.

---

### Official Review · Reviewer_okX4 · 2021-11-10

**Correctness:** 4
**Technical Novelty And Significance:** 3
**Empirical Novelty And Significance:** Not applicable
**Recommendation:** 6
**Confidence:** 3

**Main Review:**

I appreciate the rigor of the theoretical framework but admit that I have not been able to check the proofs, and therefore will comment at a heuristic level below. The notations are a bit heavy, and I would suggest that the authors consider reducing the notations in the main text, adding more discussions on comparing the error rates obtained in the different settings and by the different works, adding more intuitions for the proofs and assumptions, as well as perhaps adding a pictorial illustration of the dilated CNN model. Otherwise it can be difficult to grasp the gist of the theoretical results.

I am mainly curious about the implication of the assumption that the function has higher smoothness toward coordinates with larger indices (up to rearranging the indices). As the authors point out, this holds if the input corresponds to the Fourier coefficients and the target function is less sensitive to the higher-frequency components. However, this is not how we typically think about CNNs for image classification, where the convolutions are applied to the spatial domain rather than the frequency domain. Therefore, while the proposed model is interesting, I am not quite sure how helpful the theoretical guarantees for this scenario is for our understanding of FNNs or CNNs in actual applications. For example, I am not sure that the claim "our theoretical analysis supports the great practical success of convolutional networks" in the abstract is sufficiently convincing.

The claims that CNNs are able to “automatically extract the required index I(T, $\gamma$)” (on page 8) and “find the important indices that are relatively non-smooth compared with other indices” (on page 9) are intriguing, and I wonder if the authors could provide some further intuitive explanations behind these statements as well as about why dilation is important heuristically.

Some possible typos:
1) Page 5 top, "be back" -> "be dated back"
2) In Definition 4, if $W'$ is a number, what do ${W'}^{L'-1}, ..., {W'}^{l-1}$ mean?
3) Page 6 before Remark 5, "an dilated" -> "a dilated"
4) Page 7 before Theorem 9, "sufficiently smoothness" -> "sufficiently smooth"
5) Page 9 in Theorem 9, "there exits" -> "there exists"
6) Page 8 before Section 5.3, "As fro" -> "As for"
7) Page 9 before Section 6, "learning rates" -> "error rates" ?

**Summary Of The Paper:**

This paper studies the approximation and estimation errors of using neural networks (NNs) to fit functions on infinite-dimensional inputs that admit smoothness constraints. The authors first prove upper bounds on the approximation errors achieved by fully-connected neural networks (FNNs) that rely on certain smoothness measures of the target function, then derive more explicit bounds for using (1D diluted) convolutional neural networks (CNNs) to fit target functions that has smoothness with polynomial order increase, and finally extended the smoothness assumption to a form of sparsity constraint.

**Summary Of The Review:**

This paper provides an interesting theoretical analysis on the ability of FNNs and (a type of) CNNs to fit functions on infinite-dimensional inputs that are smooth in certain senses, though how the results may relate to the actual usage of neural networks is not completely obvious in my opinion.

---

> ### Author Response · Authors · 2021-11-22
> **Reply to reviewer okX4**
>
> Thank you for the valuable feedback which helped us improve the manuscript. Please find our answer in the following.
>
> **1. Readability**
>
> We agree that the paper is mathematically dense and require some dense notations. However, we would like to notice that the problem setting itself (nonparametric regression problem for infinite dimensional input) already requires a bit heavy functional analysis notions to rigorously deal with it. Therefore, we decided to present the contents in a rigorous way instead of giving just "intuitive" explanations to avoid any confusions.
> On the other hand, we agree that it is better that there are some pictorial illustration of proofs especially about how the dilated convolution works for feature extraction. Therefore, we have added an illustration of that in Figure 1 (Appendix G).
> Moreover, we added a table of notations in the beginning of Appendix which we consider is useful to relieve the issue of dense notations.
>
> As for the comparison to other studies, the summary of comparisons to most related studies is shown in Table 1 and additional comparisons are shown Section 3 and Appendix C.
>
>
> **2. Implication of the assumption**
>
> Indeed, the spatial dependence considered in CNN is an important aspect of CNN for vision applications. We would like to emphasize that our analysis also supports such setting. Actually, an application of our theory on the wavelet decomposition of 2D image is presented in Appendix I. The Fourier transform explanation is just one intuitive example.
>
>
> **3. I wonder if the authors could provide some further intuitive explanations behind these statements as well as about why dilation is important heuristically.**
>
> Since the dilated convolution can explore exponentially large range of inputs with respect to depth with small number of parameters, it is more efficient than using FNNs directly. Indeed, we can show that the dilated convolution can extract informative features from exponentially large range, while that FNNs requires exponentially large number of parameters to perform the same thing which deteriorates the estimation accuracy. The intuition of this effect is depicted in Figure 1 in the revised version.
>
>
> **4. Some possible typos:**
>
> Thank you very much for reading the details and pointing out the typos. We have fixed them in the revised version. The following is comments about your question:
> - In Definition 4:
> $W'$ is an integer, and $W'^{k}$ is the $k$-th power of $W'$. The superscript is not an index of matrix or tensor but represents the order of power.
> - Page 9 before Section 6, "learning rates" -> "error rates" ?:
> We have modified it to "convergence rate". More precisely, this is "convergence rate of the estimation error".

---

### Decision · Program_Chairs · 2022-01-20

**Decision:**

Accept (Spotlight)

**Comment:**

This work studies the approximation and estimation errors of using neural networks (NNs) to fit functions on infinite-dimensional inputs that admit smoothness constraints. By considering a certain notion of anisotropic smoothness, the authors show that convolutional neural networks avoid the curse of dimensionality.

Reviewers all agreed that this is a strong submission, tackling a core question in the mathematics of DL, namely developing functional spaces that are compatible with efficient learning in high-dimensional structured data. The AC thus recommends acceptance.